# Ferroptosis response segregates small cell lung cancer (SCLC) neuroendocrine subtypes

Christina M. Bebber [1,2,3], Emily S. Thomas [1,2,4], Jenny Stroh[1,2], Zhiyi Chen[1,2], Ariadne Androulidaki[1,2], Anna Schmitt[2,3], Michaela N. Höhne[2,5], Lukas Stüker[1,2], Cleidson de Pádua Alves[1], Armin Khonsari[1,6,7], Marcel A. Dammert[1,6,7], Fatma Parmaksiz[1,6,7], Hannah L. Tumbrink[1,6,7], Filippo Beleggia [3], Martin L. Sos [1,6,7], Jan Riemer[2,5], Julie George [1], Susanne Brodesser[2], Roman K. Thomas[1,6,8], H. Christian Reinhardt[9] & Silvia von Karstedt [1,2,7✉]

Loss of *TP53* and *RB1* in treatment-naïve small cell lung cancer (SCLC) suggests selective pressure to inactivate cell death pathways prior to therapy. Yet, which of these pathways remain available in treatment-naïve SCLC is unknown. Here, through systemic analysis of cell death pathway availability in treatment-naïve SCLC, we identify non-neuroendocrine (NE) SCLC to be vulnerable to ferroptosis through subtype-specific lipidome remodeling. While NE SCLC is ferroptosis resistant, it acquires selective addiction to the TRX anti-oxidant pathway. In experimental settings of non-NE/NE intratumoral heterogeneity, non-NE or NE populations are selectively depleted by ferroptosis or TRX pathway inhibition, respectively. Preventing subtype plasticity observed under single pathway targeting, combined treatment kills established non-NE and NE tumors in xenografts, genetically engineered mouse models of SCLC and patient-derived cells, and identifies a patient subset with drastically improved overall survival. These findings reveal cell death pathway mining as a means to identify rational combination therapies for SCLC.

[1] Department of Translational Genomics, Medical Faculty, University of Cologne, Cologne, Germany. [2] CECAD Cluster of Excellence, University of Cologne, Cologne, Germany. [3] Clinic I for Internal Medicine, Medical Faculty, University Hospital of Cologne, Cologne, Germany. [4] Imperial College London, London, UK. [5] Department for Chemistry, Institute for Biochemistry, University of Cologne, Cologne, Germany. [6] Institute of Pathology, Medical Faculty, University Hospital of Cologne, Cologne, Germany. [7] Center for Molecular Medicine Cologne, Medical Faculty, University Hospital of Cologne, Cologne, Germany. [8] DKFZ, German Cancer Research Center, German Cancer Consortium (DKTK), Heidelberg, Germany. [9] Department of Hematology and Stem Cell Transplantation, University Hospital Essen, University Duisburg-Essen, German Cancer Consortium (DKTK partner site Essen), Essen, Germany. ✉email: s.vonkarstedt@uni-koeln.de

Treatment of small cell lung cancer (SCLC) has remained a major clinical challenge, due to rapid relapse[1] as well as intertumoral heterogeneity which can arise through tumorigenesis from distinct cells-of-origin[2]. Moreover, Notch-driven intratumoral heterogeneity, wherein neuroendocrine (NE) SCLC cells co-occur with a non-NE chemoresistant population[3], may fuel relapse. Supporting this idea, relapsed human SCLC samples are enriched for the non-NE subtype[4]. Interestingly, recent evidence suggests that intratumoral heterogeneity in SCLC can stem from MYC-mediated induction of Notch signaling and promotion of a non-NE cellular state[5]. Moreover, intratumoral heterogeneity is further exacerbated by standard-of-care treatment making SCLC an ultimately deadly disease[6]. Molecularly, SCLC is characterized by loss of functional p53, Rb1 and a high tumor mutational burden (TMB)[7,8]. Due to the fact that SCLC presents with high TMB, a marker of potential immunotherapy response[9], clinical trials involving immune checkpoint blockade have been initiated for SCLC. Although these trials have shown encouraging results[10], they underperformed given the extent of response expected based on the very high TMB seen in SCLC[11]. A much-neglected fact is that high TMB may increase constitutive immune editing prior to treatment and thereby may select against expression of programed cell death pathway components. Interestingly, caspase 8, which is known to be essential for extrinsic apoptosis triggered by Tumor necrosis factor (TNF) superfamily ligands expressed by immune effector cells[12], has been shown to be epigenetically silenced in SCLC cell lines[13–16] suggesting this to be a strategy of immune escape. Whereas low or absent caspase 8 expression would disable the capacity to induce extrinsic apoptosis, absence of caspase 8 selectively allows for necroptosis, a non-apoptotic form of cell death driven by receptor-interacting kinase 1 (RIPK1) and RIPK3[17–19]. However, whether necroptosis is enabled in SCLC has remained unexplored. In addition, it was shown that immune checkpoint blockade and radiotherapy sensitize cancer cells to the recently described type of regulated necrosis called ferroptosis[20,21]. Thereby, ferroptosis may equally belong to the cell death arsenal triggered in tumor–immune cell interaction and selection. Ferroptosis results from an irreparable lipid peroxidation chain reaction within cellular membranes fueled by radical formation[22]. Initial oxidation of membrane lipids is facilitated by divalent iron promoting a Fenton reaction generating hydroxylradicals which react with polyunsaturated fatty acids (PUFAs)[23]. During ferroptosis, PUFAs belonging to the arachidonic acid (AA)- and adrenic acid (AdA)-containing phosphatidylethanolamine (PE) lipid species are specifically oxidized[24]. Acyl-CoA synthetase long-chain family member 4 (ACSL4) generates AA-CoA which, together with lysophospholipids, is used by lysophosphatidylcholine acyltransferase 3 (LPCAT3) as a substrate for phospholipid acylation. By generating this lipid target pool, both, ACSL4 and LPCAT3 are thereby essential determinants of ferroptosis sensitivity[25,26]. Vice versa, glutathione peroxidase 4 (GPX4) constitutively reduces accumulating lipid hydroperoxides, thereby protecting cells from ferroptosis[27,28]. GPX4 requires glutathione (GSH) as an electron donor. One important route of GSH synthesis in many cells is coupled to the availability of intracellular cysteine, which can be generated from cystine imported from the extracellular space via the cystine/glutamate antiporter System xc−. Consequently, inhibition of its small subunit xCT (SLC7A11) by the small molecule erastin results in cystine depletion and ferroptosis[29]. Recently, depletion or inhibition of xCT was shown to kill pancreatic and lung adenocarcinoma representing a novel therapeutic vulnerability in these cancers[30,31] (recently reviewed in refs. [32–34]).

The fact that SCLC almost inevitable relapses after standard-of-care treatment fueled by increased intratumoral heterogeneity

and that first-line immune checkpoint blockade only improves median survival of SCLC by two months despite high TMB[35] emphasizes the urgent need to initially understand the interplay and availability of regulated cell death pathways in this type of cancer prior to treatment. While several subtype-specific therapeutic approaches have been identified for SCLC[36–41], recent evidence suggests that SCLC subtype intratumoral heterogeneity may evolve as a consequence of subtype plasticity[5] highlighting the need to devise therapies targeting SCLC plasticity rather than isolated molecular subtypes. Here, through systematic characterization of regulated cell death pathway availability in treatment-naïve SCLC, we find broad inactivation of the extrinsic apoptosis and necroptosis pathway while we identify non-neuroendocrine (NE) and NE SCLC subtypes to mechanistically segregate by ferroptosis response. NE SCLC instead acquires selective addiction to the thioredoxin (TRX) pathway. Our data identify ferroptosis as a non-NE subtype-specific vulnerability in SCLC and suggest repurposing of combined induction of ferroptosis and TRX pathway inhibition as a strategy to address non-NE/NE SCLC intratumoral heterogeneity and prevent non-NE/NE subtype plasticity under treatment.

## Results

**Treatment-naïve SCLC presents with inactivation of regulated cell death pathways.** To profile cell death pathway availability in treatment-naïve SCLC, we analyzed RNA expression levels of a mostly treatment-naïve SCLC patient cohort in comparison with normal lung tissue samples of SCLC patients. SCLC primary patient tissue showed a strong downregulation of genes involved in the extrinsic apoptosis pathway (Fig. 1a), such as tumor necrosis factor-related apoptosis-inducing ligand (TRAIL, gene name *TNFSF10*), caspase 8 (*CASP8*), and CD95L (*FASLG*)[12] (Fig. 1d). Whereas the essential extrinsic apoptosis adaptor Fas-associated protein with death domain (*FADD*) was slightly upregulated in SCLC, potentially facilitating non-apoptotic gene induction[42,43], the downstream essential effector caspase 8 was strongly downregulated in ~80% of SCLC specimens (Fig. 1e), suggesting the extrinsic apoptosis pathway to be incapacitated prior to therapy. Indeed, a representative panel of human SCLC cell lines ($n = 7$) derived prior and post-therapy[44,45] was almost uniformly resistant to extrinsic apoptosis triggered by TRAIL, which killed a TRAIL-sensitive control non-small cell lung cancer (NSCLC) cell line (Fig. 1f). Downregulation of *CASP8*, can enable unrestricted activation of the necroptosis pathway through relieving *CASP8*-imposed negative regulation via *RIPK1* and *3*[17,18] (Fig. 1b). Therefore, we next performed a cluster analysis of necroptosis pathway components (Fig. 1g). Yet, while *RIPK3* was slightly upregulated hinting at potential tumor-promoting functions[46,47], expression of the downstream essential necroptosis effector mixed lineage kinase domain like pseudokinase (*MLKL*)[48–50] was strongly downregulated in SCLC, as compared to normal lung (Fig. 1h). Accordingly, SCLC cell lines were resistant to experimental induction of necroptosis by TNF, zVAD, and smac mimetic (TZS), in contrast to mouse embryonic fibroblast (MEFs) control cells which readily underwent necroptosis upon TZS stimulation (Fig. 1i; Supplementary Fig. 1a). Importantly, TZS-induced cell death was blocked by co-treatment with the RIPK1 inhibitor Necrostatin-1s (Nec-1), thereby confirming the induction of necroptotic cell death in MEFs (Fig. 1i). These data suggest that strong selective pressure may already incapacitate both, the extrinsic apoptosis and necroptosis pathways in SCLC prior to diagnosis and therapy.

Recently, ferroptosis, an iron-dependent form of regulated necrosis was described[51,52]. Importantly, ferroptosis is independent of the molecular machinery driving apoptosis or necroptosis

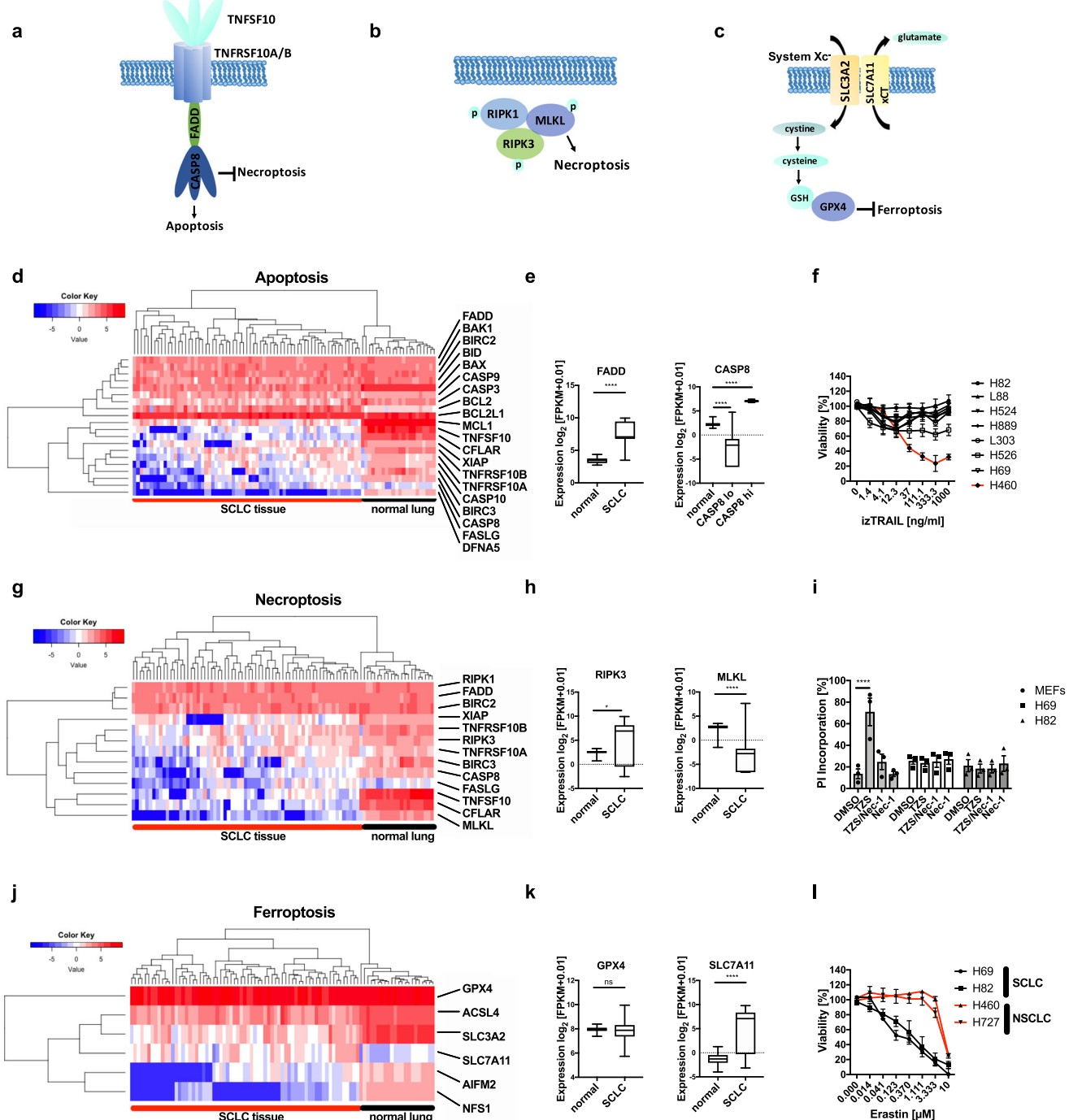

**Fig. 1 Regulated cell death pathways are counter-selected in treatment-naïve SCLC. a** Schematic view of genes involved in extrinsic apoptosis, **b** necroptosis and **c** ferroptosis. **d, e, g, h, j, k** RNA-seq expression data in FPKM (fragments per kilobase of exon model per million reads mapped) from normal lung[85] ($n = 22$) and mostly chemo-naïve SCLC patient samples[7] ($n = 67$) were log2 transformed ($+0.01$) and plotted for relative expression of genes involved in extrinsic apoptosis (**d, e**) necroptosis (**g, h**) and ferroptosis (**j, k**) boxplot center line, mean; box limits, upper and lower quartile; whiskers min. to max. Heatmap color code indicates expression levels between each sample and the average of each gene, dendrogram shows the distance between sample populations. **f** The indicated seven human SCLC cell lines ($n = 7$) and NSCLC cell line (H460) were treated with human izTRAIL for 24 h, cell viability was determined by Cell Titer Blue, $n$ (H82) = 6, $n$ (L88) = 4, $n$ (H524) = 4, $n$ (H889) = 4, $n$ (L303) = 3, $n$ (H526) = 3, $n$ (H69) = 5, $n$ (H460) = 3. **i** Cells were treated with TNF (T) [10 ng/ml]/zVAD (Z) [20 µM]/Smac mimetic (S) Birinapant [1 µM] $+/-$ Nec-1 [10 µM] for 24 h, cell death was quantified by propidium iodide (PI) uptake and flow cytometry. **l** Cells were treated with erastin at indicated concentrations for 24 h, cell viability was determined by Cell Titer Blue. Data are means ± SEM of three independent experiments wherever not indicated otherwise. Two-way ANOVA + Tukey's multiple comparison test (**i**) and two-tailed unpaired $t$ tests for all others, ****$p < 0.0001$, *$p < 0.05$, ns $p > 0.05$. Source data are provided as Source data file.

(Fig. 1c). Therefore, we next assessed expression of ferroptosis regulators in SCLC patient samples. Strikingly, *SLC7A11* was strongly upregulated in SCLC and only expressed at very low levels in normal lung (Fig. 1j, k). *GPX4* was highly and comparably expressed in SCLC and normal lung. Moreover, in human SCLC cell lines, protein expression of both xCT and GPX4 could be validated (Supplementary Fig. 1b). Elevated xCT expression might therefore represent a selective advantage for SCLC—as opposed to normal lung—exposing a potential SCLC-selective therapeutic opportunity. Confirming this notion, treatment of SCLC cells, but much less of NSCLC cells, with the xCT inhibitor erastin resulted in dose-dependent cytotoxicity (Fig. 1l). Moreover, when directly comparing erastin sensitivity of murine cells from genetically engineered mouse models for SCLC[53] (RP-mice) and NSCLC[54] (KP-mice) both on the C57BL/6 strain background, again SCLC cells were overall more sensitive to erastin-induced cytotoxicity (Supplementary Fig. 1c). While GPX4, xCT and ACSL4 levels were expressed at comparable levels between SCLC and NSCLC cells, interestingly, transferrin receptor (CD71) expression was elevated in all SCLC cell lines as compared to two out of three NSCLC lines. Moreover, expression of ferroptosis suppressor protein 1 (FSP1, formerly AIFM2), recently demonstrated to render cells more resistant to ferroptosis[55,56], was elevated specifically in NSCLC (Supplementary Fig. 1d) cells suggesting that superior intracellular iron availability combined with low FSP1 expression may specifically prime SCLC for ferroptosis induction. In addition, viability data taken from an erastin screen in 117 cancer cell lines[28] independently confirmed increased erastin sensitivity of SCLC, over comparable NSCLC cell lines (Supplementary Fig. 1e). Taken together, we find that SCLC has evolved to present with cell death resistance on multiple levels prior to treatment, i.e. against extrinsic apoptosis, necroptosis, and ferroptosis. However, since ferroptosis escape in SCLC, unlike escape from other cell death pathways, involves the upregulation of protective and targetable proteins, rather than the loss of agonists, we next aimed to mechanistically validate the induction of ferroptosis in SCLC.

**SCLC is vulnerable to the induction of ferroptotic cell death**. To validate whether SCLC was vulnerable to ferroptotic cell death in general, murine SCLC cells were treated with the GPX4 small molecule inhibitor RSL3, known to trigger ferroptosis[28] with or without the lipophilic radical scavenger Ferrostatin-1 (Fer-1) shown to specifically block ferroptotic cell death[51]. Indeed, RSL3 and erastin effectively abrogated clonogenic survival of adherent murine SCLC lines, which could be fully restored with Fer-1 co-treatment in the case of RSL3 and partially rescued in the case of erastin, which is known to also affect other pathways (Fig. 2a). Moreover, additional cell death induced by erastin or two structurally distinct small molecule inhibitors against GPX4 (RSL3[57] and ML210[58]) could be rescued by co-incubation with Fer-1 (Fig. 2b, c; Supplementary Fig. 2a, b) in human SCLC cells, which contained some non-specific background cell death due to culture in suspension. Co-treatment with Fer-1 blocked RSL3-induced cell death across a range of different doses (Supplementary Fig. 2c). Using the fluorescent probe Calcein-AM, which can be used to detect relative amounts of the intracellular labile iron pool through a principle of fluorescent quenching[59], we first confirmed iron depletion by the iron-scavenger deferoxamine (DFO) which indeed led to a loss-of fluorescent quenching (Supplementary Fig. 2d). DFO equally rescued cell death induced by erastin or GPX4 inhibition (Fig. 2d, e). Lastly, cell death induced by erastin or RSL3 in murine SCLC cell lines could also be rescued by Fer-1 (Supplementary Fig. 2e, f). Together, these data indicate that RSL3, ML210 and erastin induces lipid ROS- and iron-dependent cell

death in SCLC - hallmarks of ferroptosis. While GPX4 is central to protecting a variety of cells from ferroptosis, in lung adenocarcinoma cells, *GPX4* deletion is not sufficient to induce ferroptosis[56]. Therefore, to determine whether or not GPX4 is sufficient to protect SCLC from ferroptosis, we generated CRISPR/Cas9-mediated control or *GPX4* knockout (KO) SCLC lines. Supporting a vital function for GPX4 in preventing toxic build-up of lipid reactive oxygen species (ROS) in SCLC, these cells could only be generated and cultured in the presence of Fer-1 in the media. Consequently, upon Fer-1 withdrawal, cells with *GPX4*-targeting gRNAs selectively died (Fig. 2f, Supplementary Fig. 2g). GPX4 KO was confirmed on protein level in the presence of Fer-1 (Fig. 2g). In addition, GPX4 KO SCLC cells presented with lipid ROS accumulation upon Fer-1 withdrawal, indicative of the induction of ferroptotic cell death[51] (Fig. 2h, i, Supplementary Fig. 2h). Collectively, these data support a requirement for free iron and lipid radicals in cell death execution upon pharmacological induction of ferroptosis and appoint GPX4 as the central player sufficient to prevent lipid peroxidation and ferroptosis in SCLC.

**Non-NE SCLC is exquisitely sensitive to ferroptosis**. SCLC can be subdivided into several molecular subtypes based on patterns of expression of neuroendocrine (NE) differentiation markers, which differ in their cell biology[3], cell of origin or arise as a consequence of intratumoral plasticity[5]. Therefore, we next aimed to understand whether ferroptosis sensitivity was a common feature of all molecular subtypes of SCLC. To address this, we tested ferroptosis sensitivity in a larger panel of human ($n = 8$) and murine ($n = 6$) SCLC lines representative of described SCLC molecular subtypes[1]. Interestingly, we noted that this panel divided into subsets of responders (black) and non-responders (gray) in response to all inducers of ferroptosis tested (Fig. 3a, b; Supplementary Fig. 3a–c). This separation was also reflected in failure to accumulate lipid ROS upon GPX4 inhibition in human and murine non-responders (Fig. 3c, Supplementary Fig. 3d). Moreover, while human non-responders were still capable to accumulate total ROS upon induction of ferroptosis, murine non-responders also did not accumulate total ROS upon induction of ferroptosis (Supplementary Fig. 3e, f). Thereby, human and murine ferroptosis non-responders were specifically incapacitated to accumulate lipid ROS. Of note, this divided response pattern was unique to ferroptosis, as the same murine and human SCLC cell line panels did not show a bipartite response to cisplatin or etoposide (Supplementary Fig. 4a–d). Cisplatin, as well as etoposide, instead led to caspase-dependent cell death as pan-caspase inhibitor zVAD partially reverted cell death induction (Supplementary Fig. 4e, f).

To identify markers of ferroptosis sensitivity and/or resistance in SCLC responders and non-responders, we analyzed mRNA expression patterns of human SCLC cells[36], comparing ferroptosis responders and non-responders which clustered into two distinct groups (Fig. 3d). Interestingly, while non-responders showed increased expression of the transcription factor Achaete-Scute Homolog 1 (*ASCL1*), responders instead expressed elevated mRNA levels of RE1-silencing transcription factor (*REST1*, also known as neuron-restrictive silencing factor NRSF) (Fig. 3e). In line with this, gene set enrichment analysis (GSEA) revealed that upregulated ASCL1 target genes[60] were enriched in non-responders while a REST1 signature was enriched in ferroptosis sensitive cells (Fig. 3f). While ASCL1 is known to promote NE differentiation in SCLC, REST1 suppresses this process. Intriguingly, mouse SCLC cells can spontaneously transdifferentiate from non-NE (REST1+, ASCL1−) to an NE-ASCL1+ (NE) state phenotypically marked by adherent growth (stickers) and growth in suspension (floaters), respectively[3]. Making use of this system of isogenic spontaneous NE differentiation, we generated murine

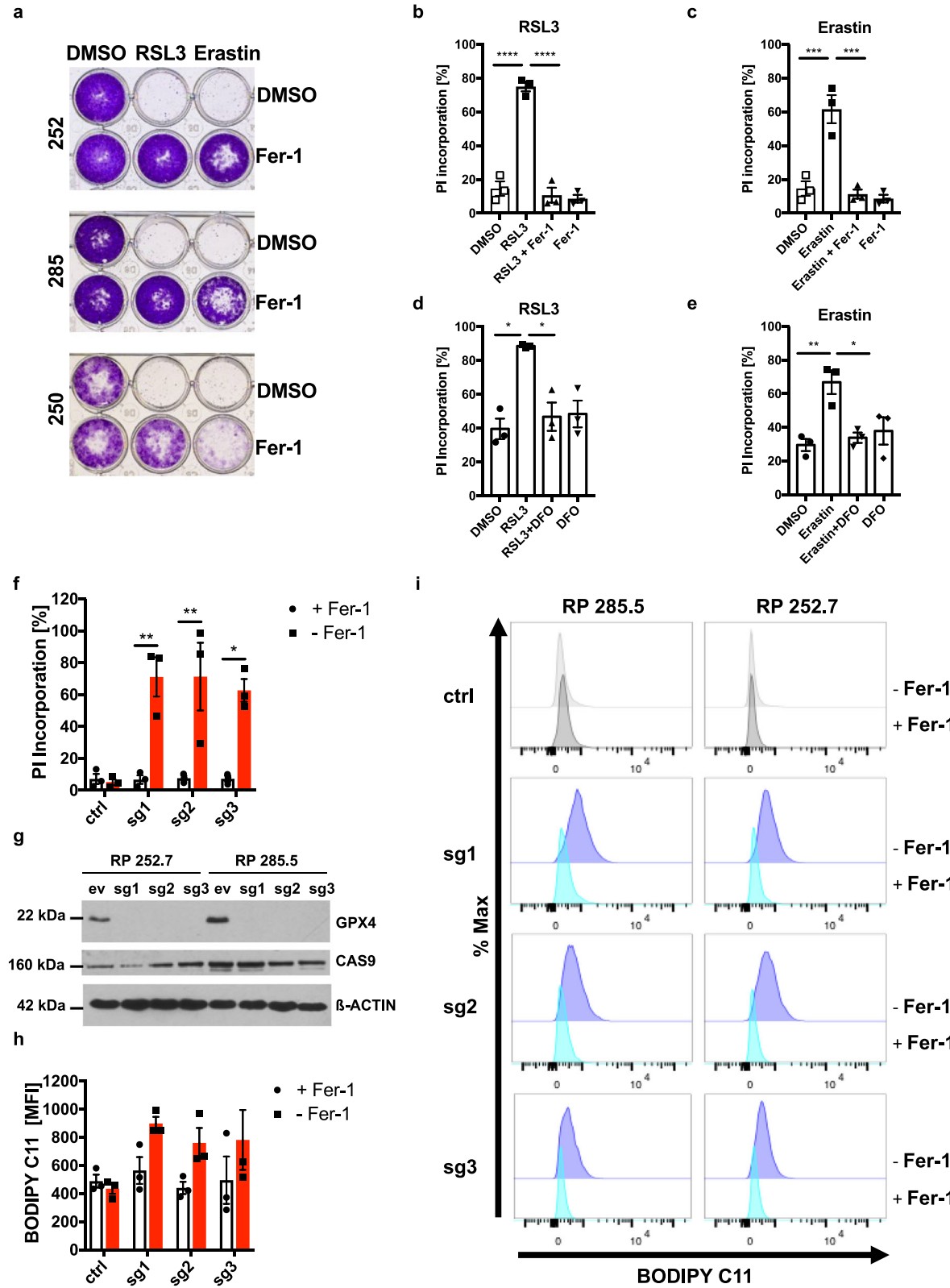

SCLC cell lines from three different tumor-bearing RP-mice and validated spontaneous transition between a stickers and floaters state in these cells (Supplementary Fig. 5a). Moreover, we validated expression of non-NE (REST1, YAP1, vimentin (VIM)), and NE protein marker expression (ASCL1, synaptophysin (SYP), neural cell adhesion molecule (NCAM)) in these three isogenic pairs of stickers and floaters, respectively (Fig. 3g). Strikingly, all isogenic NE floaters were indeed more resistant to ferroptosis triggered by GPX4 inhibition, or erastin, while non-NE stickers were highly sensitive (Fig. 3h; Supplementary Fig. 5b). Moreover, floaters unlike stickers showed impaired accumulation of lipid ROS and total ROS upon GPX4 inhibition (Supplementary Fig. 5c, d).

**Fig. 2 GPX4 expression is sufficient to protect SCLC from ferroptosis. a** The indicated murine SCLC lines ($n = 3$) were treated either with DMSO, RSL3 [1 µM], erastin [10 µM] alone or in combination with Ferrostatin-1 (Fer-1) [5 µM] for 24 h. Cells were washed, cultured for 5 days for recovery and stained with crystal violet. **b–e** Human H82 cells were treated as indicated: erastin [10 µM] ± Ferrostatin-1 (Fer-1, 5 µM) ± deferoxamine (DFO) [100 µM], RSL3 [1 µM] ± Fer-1 ± DFO for 24 h. Cell death was determined by propidium iodide (PI) uptake and flow cytometry. **f** RP285.5 murine SCLC cells stably expressing Cas9 and the indicated control or gRNAs targeting *GPX4* were cultured with or without Ferrostatin-1 (Fer-1) for 24 h. Cell death was quantified by propidium iodide (PI) uptake and flow cytometry. **g** Protein extracts were obtained from cells as in (**f**) cultured in the presence of Fer-1 [5 µM]. A representative western blot is shown (**h, i**) RP285.5 murine SCLC cells as in (**f**) were cultured in the presence or withdrawal of Fer-1 [5 µM] for 5 h and stained for lipid ROS accumulation using BODIPY C11. Cells were analyzed by flow cytometry and mean fluorescent intensity (MFI) was quantified. Data are means ± SEM of three independent experiments in each individual cell line or representative images or histograms were applicable. One-way ANOVA + Tukey's multiple comparison test (**b–e**), Two-way ANOVA + Tukey's multiple comparison test (**f**), ****$p < 0.0001$, ***$p < 0.001$, **$p < 0.01$, *$p < 0.05$. Source data are provided as Source data file.

---

Again, this segregation of cell death sensitivity was specific to ferroptosis, as stickers and floaters responded equally to cisplatin or etoposide treatment (Supplementary Fig. 5e, f). Moreover, cell adherent or non-adherent states, in general, did not determine ferroptosis sensitivity or resistance, as all human cell lines tested, including responders and non-responders, grow in suspension (Fig. 3a). Recently, cMyc was shown to promote transition to a non-NE SCLC phenotype which acquires high Yes-associated protein 1 (YAP1) expression over time[5,36]. Therefore, to validate ferroptosis response segregation by NE and non-NE differentiation in an independent and genetically-defined experimental cellular set-up, we made use of SCLC cells derived from the RP mouse model in which Adenoviral-Cre-mediated deletion of Rb1 and Tp53 in the lung gives rise to NE ASCL1+ SCLC as compared to cells derived from the RPM mouse model in which additional expression of cMyc$^{T58A}$ leads to largely non-NE differentiated tumors[36]. Strikingly, RPM cells were significantly more sensitive to ferroptosis than RP cells and indeed showed slightly increased expression of the non-NE markers REST1 and VIM (Fig. 3j, k; Supplementary Fig. 5g). Moreover, cMyc expression induced from the endogenous locus via CRISPR activation (CRISPRa)[37] in RP cells increased ferroptosis sensitivity in comparison to control cells (Supplementary Fig. 5h, i). Importantly, RP, RPM, and cMyc CRISPRa cells all grow under adherent conditions confirming that differences in ferroptosis sensitivity in SCLC cells do not stem from growth under non-adhesion/adhesion conditions. Interestingly, cMyc$^{T58A}$ expression was shown to promote gradual non-NE differentiation in SCLC upregulating YAP1 in the process[5]. Moreover, YAP1 was recently shown to promote ferroptosis sensitivity at low cellular confluence[61]. Therefore, we tested whether expression of a constitutively active mutant of YAP1 in which all five serines have been mutated to alanines (YAP1 5SA) was sufficient to render non-NE stickers even more ferroptosis sensitive. Indeed, overexpression of YAP1 5SA rendered stickers even more sensitive to ferroptosis while also increasing expression of the non-NE marker REST1 (Fig. 3k; Supplementary Fig. 5j). These data propose that ferroptosis sensitivity is gradually acquired along stages of NE to non-NE differentiation.

**Non-NE/NE SCLC subtypes are characterized by lipid metabolism remodeling.** During ferroptosis, characteristic types of PUFAs, including AA and AdA lipid species which are generated by ACSL4 and LPCAT3 have been shown to be specifically peroxidized[24]. Therefore, we first performed RNA-sequencing of representative 181.5 stickers as compared to floaters to compare expression of common ferroptosis regulators, including ACSL4 and LPCAT3. While expression patterns of other ferroptosis regulators (GPX4, xCT, FSP1/AIFM2, CD71/TFRC) could not explain increased ferroptosis sensitivity of non-NE stickers, ACSL4 and LPCAT3 were indeed expressed at slightly elevated levels (Supplementary Fig. 5k). Independently confirming this correlation, in the NIH SCLC cell line panel, ASCL1 mRNA

expression also inversely correlated with ACSL4 expression (Supplementary Fig. 5l) proposing increased availability of PE peroxidation target lipids to facilitate ferroptosis in non-NE SCLC. Therefore, we next assessed how spontaneous NE differentiation in isogenic stickers as compared to floaters may affect the oxidized lipidome upon induction of ferroptosis using mass spectrometry. Strikingly, already basal ferroptosis-specific phospholipid peroxidation products, including oxidized PUFA-PE species, were elevated in stickers and further increased upon RSL3 treatment, while both, the basal amounts and the specific induction of these products, were markedly decreased in floaters (Fig. 4a and Supplementary Data 1). Given that already basal amounts of oxidized PUFAs markedly differed between stickers and floaters we hypothesized that non-NE/NE transdifferentiation may affect the PUFA-generating phospholipid metabolism. To test this, we performed mass spectrometry analysis of total phospholipids in stickers and floaters. Interestingly, while the majority of diacylglycerol (DAG) PUFA levels detected were either comparable or elevated in floaters (Fig. 4b, c), stickers showed selective upregulation of several ether-linked PUFA species (Fig. 4d, e). Therefore, we next determined mRNA expression of enzymes involved in the specific synthesis of these ether-linked PUFAs as well as DAG PUFAs in our panel of isogenic non-NE stickers as compared to NE-floaters. Strikingly, there was a clear trend of several of these enzymes to be expressed at higher levels in non-NE SCLC cells explaining the elevated levels of ether-linked PUFAs detected. Yet, also enzymes specifically involved in AA incorporation into phospholipids, including LPCAT3 and ACSL4, were upregulated in stickers (Fig. 4f), suggesting lipid metabolism remodeling during non-NE/NE transdifferentiation to contribute to ferroptosis sensitivity and resistance, respectively. Since ether lipid species were specifically elevated in stickers and ether-linked PUFAs and their synthesis were recently shown to promote ferroptosis sensitivity[62], we hypothesized that expression of ether lipid synthesis enzymes and with that increased ether-linked PUFA synthesis contributes to elevated ferroptosis sensitivity of non-NE SCLC. Strikingly, upon silencing of the ER-resident ether lipid synthesis enzyme 1-acylglycerol-3-phosphate O-acyltransferase 2 (*AGPAT2*) or *AGPAT3*, non-NE SCLC cells were indeed less sensitive to ferroptosis (Fig. 4g, h; Supplementary Fig. 5m, n).

Lastly, to determine whether elevated expression of ether lipid synthesis pathway components in non-NE SCLC would also hold true in treatment-naïve patients, we clustered patients as compared to normal lung according to *ASCL1* expression. Although the treatment-naïve SCLC patient dataset only contains 10 *ASCL1*$^{low}$ patients, there was a visible trend for an inverse correlation of *GNPAT*, *ACSL4* and, to a lesser extent, *LPCAT3* expression with *ASCL1* expression (Supplementary Fig. 6). Taken together, we find downregulation of ether lipid metabolism to be a hallmark of non-NE/NE transdifferentiation and increased ether lipid synthesis to promote ferroptosis sensitivity in non-NE

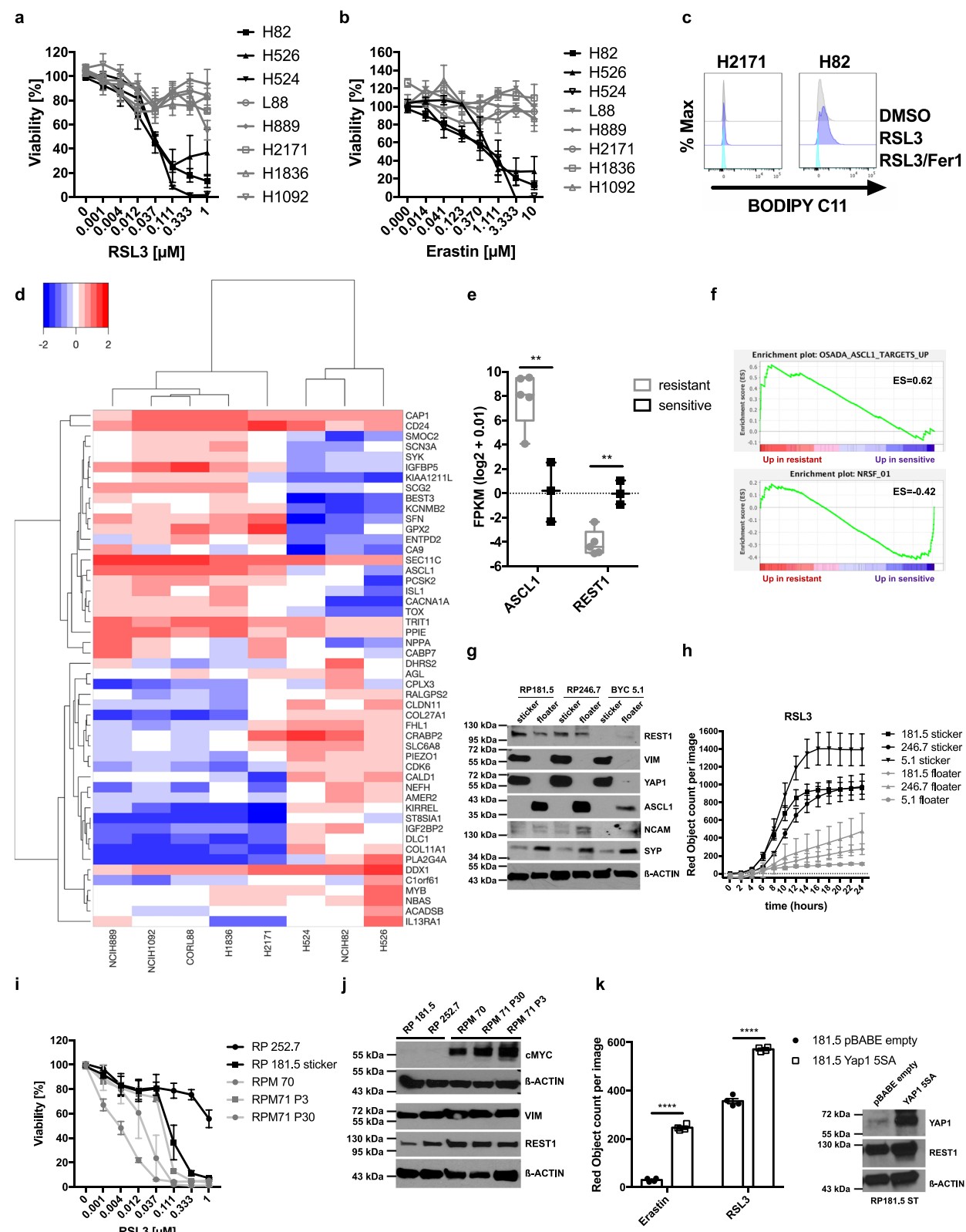

SCLC. Thereby, non-NE/NE plasticity in SCLC involves down-regulation of ether lipid synthesis resulting in a lipidome less vulnerable to lipid peroxidation.

**NE SCLC is defined by selective addiction to the TRX anti-oxidant pathway.** While we found non-NE SCLC to be ferroptosis sensitive due to upregulated ether lipid synthesis, NE SCLC was highly resistant. In order to determine whether NE SCLC may have acquired ferroptosis resistance through upregulating GSH and thereby GPX4 activity, we measured cellular concentrations of GSH and its oxidized form GSSG. Surprisingly and contrary to expectations, NE floaters and human and murine resistant NE

**Fig. 3 SCLC neuroendocrine subtypes segregate by ferroptosis response. a, b** Eight human SCLC cell lines were treated as indicated for 24 h, cell viability was determined by Cell Titer Blue. **c** The indicated human SCLC cells were treated with DMSO, RSL3 [100 nM] or RSL3/Fer-1 [5 μM] for 5 h and stained for lipid ROS accumulation using BODIPY C11. Cells were analyzed by flow cytometry. Gray = DMSO treated, violet = RSL3 treated, turquoise = RSL3/Fer1 treated. **d** RNA-seq data[36] of human SCLC lines (sensitive n = 3 cell lines, H524, NCIH82, H526; resistant n = 5 cell lines, NCIH889, NCIH1092, CORL88, H1836, H2171) were analyzed for differential expression between responders and non-responders, heatmap represents hierarchical clustering of FPKM (log2 + 0.01) of the 50 most differentially expressed genes. Heatmap color code indicates expression levels between each sample and the average of each gene, dendrogram shows the distance between sample populations. **e** *ASCL1* and *REST1* expression (FPKM (log2 + 0.01) comparing three sensitive and five resistant cell lines is plotted, boxplot center line, mean; box limits, upper and lower quartile; whiskers min. to max. **f** Gene set enrichment analysis (GSEA) of a ranked list from ferroptosis sensitive and resistant cells was performed. **g** Western blot of SCLC NE subtype marker expression in the indicated manually separated stickers and floaters lines (n = 3). **h** Manually separated stickers and floaters lines (n = 3) were treated with RSL3 [1 μM] for 24 h. DRAQ7 [0.1 μM] was added to all wells to visualize dead cells. Images were acquired every 2 h using the IncuCyte S3 bioimaging platform. **i** The indicated mouse RP and RPM cell lines were treated with rising concentrations of RSL3 for 24 h, cell viability was determined by Cell Titer Blue. **j** Cells as in (**i**) were analyzed for protein expression by western blotting. Representative western blots are shown **k** 181.5 stickers expressing either vector control or YAP1 5SA-YFP were treated with RSL3 [1 μM] for 24 h. DRAQ7 [0.1 μM] was added to all wells to visualize dead cells. Images were acquired every 2 h using the IncuCyte S3 bioimaging platform, n = 4 biological replicates. Representative western blots are shown. Data are means ± SEM of three or more independent experiments or representative images if not indicated otherwise. Two-tailed unpaired t tests, ****p < 0.0001, **p < 0.01. Source Data are provided as Source data file.

cells, in fact, presented with lower basal levels of GSH whilst levels of reduced GSSG were comparable with those in responders and stickers (Supplementary Fig. 7a, b). These data indicated that GSH synthesis but not recovery is specifically repressed in ASCL1-expressing NE SCLC. In support of this, inducible expression of ASCL1 suppressed expression of glutamate-cysteine ligase catalytic subunit (GCLC) an essential enzyme in the synthesis of GSH (Supplementary Fig. 7c). Consequently, inducible expression of ASCL1 also suppressed cellular amounts of GSH (Supplementary Fig. 7d). Yet, in line with prior observations in SCLC non-NE/NE transition[3], inducible expression of ASCL1 in non-NE stickers was insufficient to induce an NE differentiated floater phenotype. Hence, inducible expression of ASCL1 in stickers was also insufficient to render cells more resistant to ferroptosis (Supplementary Fig. 7e). Therefore, ASCL1 expression can serve as a marker of ferroptosis resistance in fully NE differentiated human SCLC, yet its isolated expression is insufficient to drive cell fate switch in our cellular system and along with that ferroptosis resistance. Nevertheless, ASCL1 expression acutely lowers the cellular redox potential endowed by cellular GSH. Yet, all fully NE differentiated SCLC cells tested were more resistant to ferroptosis induced by RSL3, erastin and also cystine starvation (Supplementary Fig. 7f). These data suggested that this initial collapse of GSH mediated anti-oxidant defense upon ASCL1 expression must drive a compensatory switch to usage of an entirely GSH-independent anti-oxidant defense system in NE SCLC.

Interestingly, GCLC-deficiency can be compensated for by the thioredoxin (TRX) anti-oxidant pathway and vice versa[63,64]. As part of the thioredoxin (TRX) anti-oxidant pathway, thioredoxin reductase 1 (TrxR1, gene name TXNRD1) reduces oxidized TRX which can then function to reduce other oxidized cellular substrates. Reduced TRX can, in turn, be sequestered by TXNIP limiting TRX availability for anti-oxidant defense (Fig. 5a). Given that we found ASCL1 expression to suppress GCLC in NE SCLC, we hypothesized that NE SCLC may become selectively dependent on the TRX pathway for anti-oxidant defense. Therefore, we monitored protein expression of TRX pathway components comparing human non-NE with NE cells as well as stickers with floaters. Interestingly, while TrxR1 and TrxR2 were upregulated only in human NE SCLC cells, expression of TXNIP was consistently increased in human and murine NE cells suggesting NE SCLC cells to also experience a limited availability TRX-mediated anti-oxidant defense (Fig. 5b; Supplementary Fig. 7g). Indeed, treatment with the TrxR1 and TrxR2 inhibitor Auranofin—clinically approved for the treatment of rheumatoid arthritis—led to a quicker time-dependent loss of fully-reduced

TRX in NE as compared to non-NE SCLC cells supporting pre-existing inhibition of the TRX pathway in NE-A SCLC (Fig. 5c; Supplementary Fig. 7h). Strikingly, this translated into selective sensitivity to Auranofin-induced cell death in NE SCLC while non-NE SCLC was more resistant to Auranofin (Fig. 5d). The same NE-selective effect could be observed for two other structurally distinct inhibitors of TrxR1 (PX 12 and D9) (Supplementary Fig. 7i, j). Vice versa, the same set of cell lines showed an inverse response pattern to ferroptosis induced by the clinically advanced GCLC inhibitor buthionine sulfoximine (BSO) (Supplementary Fig. 7k). Importantly, BSO triggered Ferrostatin-1 blockable ferroptosis in non-NE stickers, while Auranofin-induced cell death in NE floaters was non-ferroptotic (Fig. 5e, f). Confirming induction of ferroptotic cell death in stickers, BSO also induced lipid ROS accumulation which was reversed by Ferrostatin-1 co-treatment (Supplementary Fig. 7l). Therefore, NE SCLC is selectively addicted to the TRX anti-oxidant pathway while non-NE is sensitive to ferroptosis, a concept which holds true in a panel of isogenic cells spontaneously undergoing non-NE/NE plasticity (Supplementary Fig. 8a).

SCLC tumors are known to present with intratumoral heterogeneity in respect to individual cellular NE differentiation[3,36]. Moreover, this NE subtype intratumoral heterogeneity can originate from plasticity[5]. Therefore, to mimic a situation of NE intratumoral heterogeneity, we next determined how induction of either ferroptosis or TRX pathway inhibition would affect an isogenic mixed ASCL1+/ASCL− (non-NE/NE) culture. For this, we made use of the fact that the manually separated stickers culture already contains a subpopulation of cells in the process of NE transdifferentiation with high expression of ASCL1. Strikingly, while a morphologically distinct subpopulation of SCLC cells with bright nuclear ASCL1 staining could be detected by microscopy in control-treated cells, this population was selectively depleted upon treatment with Auranofin which did not affect numbers of ASCL1− cells as visualized by DAPI. Vice versa, induction of ferroptosis using BSO led to a strong relative enrichment in ASCL1+ cells while the majority of ASCL1− cells was killed (Fig. 5g). Strikingly, when gating only on live cells in mixed sticker/floater cultures, a proportion of surviving stickers transdifferentiated to NE SCLC by acquiring ASCL1 expression under BSO treatment. Surviving floaters instead lost ASCL1 expression under Auranofin treatment revealing a proportion of non-NE/NE plasticity to depend on cellular redox signaling in SCLC (Fig. 5h). Together, these data suggest that in heterogeneous non-NE/NE SCLC tumors,

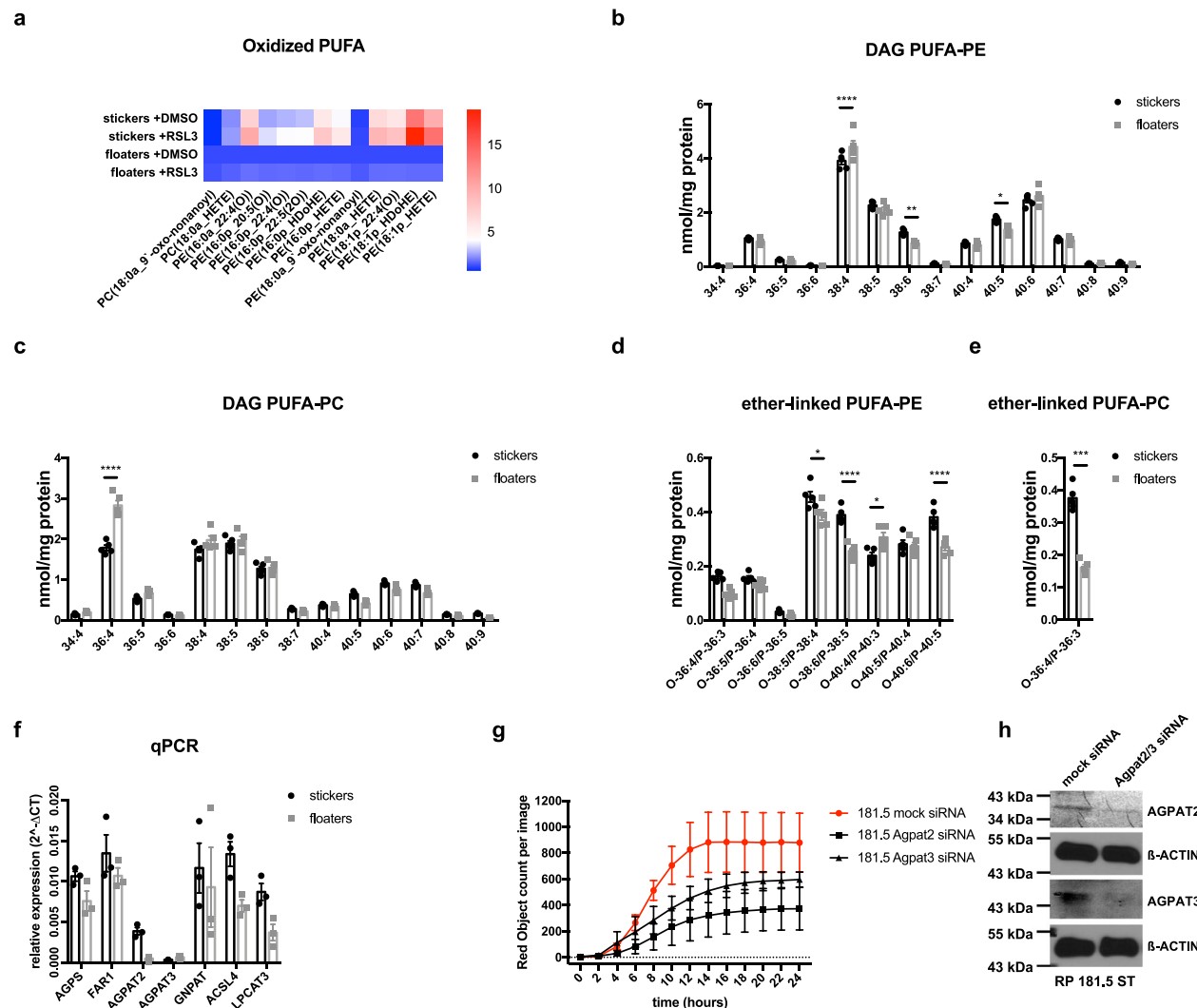

**Fig. 4 non-NE/NE SCLC subtypes undergo lipid metabolism remodeling. a** Heatmap showing the representation of mono-oxidized phospholipid species (PE phosphatidylethanolamine; PC phosphatidylcholine) in 181.5 stickers as compared to 181.5 floaters treated with either DMSO or RSL3 [1 µM] for 5 h and then subjected to lipidomics. Samples for each condition ($n = 5$) were averaged and normalized to the cell number ($2.5 \times 10^6$). Each lipid species was normalized to levels detected in floaters +DMSO. One representative out of two independent experiments is shown. Heatmap color code indicates normalized lipid species levels of each sample. **b–e** 181.5 stickers floaters ($n = 5$ samples) as compared to 181.5 floaters ($n = 5$ samples) were analyzed for basal diacylglycerol (DAG) and ether-linked lipids by mass spectrometry. Lipid content was normalized to infused protein for each condition and replicate. Individual PUFAs (4 double bonds or more) are plotted. **f** RNA was isolated from three stickers/floaters lines (RP181.5; RP246.7; BYC5.1), respective cDNA obtained and qPCR performed for the indicated transcripts. **g** RP181.5 were subjected to the indicated siRNA-mediated knockdowns for 72 h and then treated with RSL3 [1 µM] for an additional 24 h. DRAQ7 [0.1 µM] was added to all wells to visualize dead cells. Images were acquired every 2 h using the IncuCyte S3 bioimaging platform. Dead cells/image are normalized to cell confluence at the beginning of RSL3 treatment. **h** Representative western blots are shown. Data are means ± SEM of three independent experiments or representative images if not indicated otherwise. Two-tailed unpaired $t$ tests (**e**) Two-way ANOVA + Tukey's multiple comparison test, ****$p < 0.0001$, ***$p < 0.001$, **$p < 0.01$, *$p < 0.05$. Source data are provided as Source data file.

fractional killing via ferroptosis may select for NE cells over time while, vice versa, TRX pathway inhibition might enrich tumors for non-NE SCLC cells. Moreover, single pathway targeted treatment-induced plasticity will equally enable escape of SCLC subtypes by transdifferentiation. To pre-empt fractional killing and selection as well as non-NE/NE plasticity under treatment, we evaluated the extent of synergistic killing by combined BSO/Auranofin in non-NE and NE SCLC cells. Indeed, combined BSO/Auranofin treatment induced very effective cell death in both subtypes (Supplementary Fig. 8b, c). When combining sublethal doses of Auranofin with BSO, cell death could only be reverted by combined Ferrostatin-1 and N-acetylcysteine treatment (NAC) (Supplementary Fig. 8d, e), the latter of which is a general antioxidant with poor efficiency against ferroptosis[51,65].

These data indicated that cell death induced by this treatment combination in SCLC was partially dependent on lipid ROS (blocked by Ferrostatin-1) and, thereby, partially ferroptotic and partially dependent on general ROS (blocked by NAC). Confirming target specificity of this regime in SCLC, only combined siRNA-mediated suppression of GCLC, TrxR1, and TrxR2, led to an equal extent of specific cell death induction in stickers and floaters (Supplementary Fig. 8f, g). In line with this, knockdown of xCT alone was insufficient to kill non-NE SCLC. However, xCT suppression also showed a trend toward sensitization for Auranofin (Supplementary Fig. 8h, i). These data suggested very rapid redox pathway plasticity enabling ferroptosis escape within the 48 h window of the knockdown via alternative use of the TRX pathway for anti-oxidant defense.

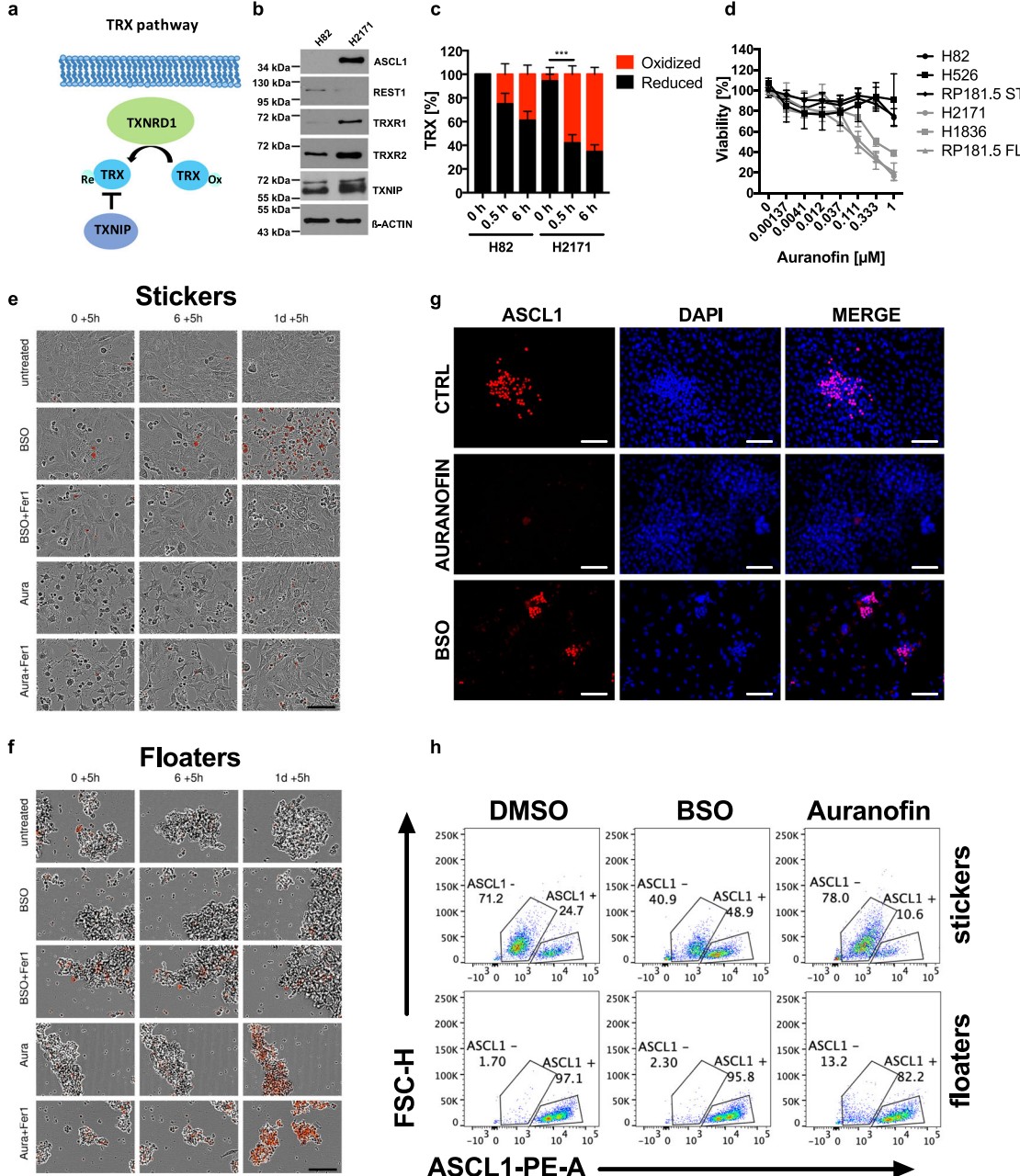

**Fig. 5 Neuroendocrine SCLC presents with TRX pathway addiction. a** Schematic view of genes involved in the TRX anti-oxidant pathway. **b** H82 and H2171 cells were lyzed and TRX pathway component expression was detected by western blot. Representative western blots are shown. **c** Indicated cells were treated with Auranofin [1 μM] for the indicated times and subjected to redox shift assays. Densitometrical quantification of TRX redox forms is shown. **d** Indicated cells were treated with Auranofin for 24 h, cell viability was determined by Cell Titer Blue. **e**, **f** RP181.5 manually separated stickers and floaters were treated with Ferrostatin-1 [5 μM] for 2 h prior to adding DMSO, BSO [10 mM] or Auranofin [1 μM] for an additional 24 h. DRAQ7 [0.1 μM] (red color in image) was added to all wells to visualize dead cells. Images were acquired every 5 h using the IncuCyte S3 bioimaging platform. Scale bar = 100 μm. **g** RP181.5 stickers were treated with either DMSO, Auranofin [500 nM] or BSO [500 μM] for 96 h and then fixed and stained for ASCL1 (red) and counterstained with DAPI (blue). Scale bar = 100 μm. **h** Manually separated stickers and floaters were treated with either DMSO, BSO [10 mM] or Auranofin [1 μM] for 24 h. Cells were gated on live cells and analyzed for ASCL1 expression by flow cytometry. FSC-H, forward scatter-heights. Data are means ± SEM of three independent experiments or two (**c**) or representative images out of at least two independent experiments are shown. Two-way ANOVA + Tukey's multiple comparison test, ***$p < 0.001$. Source data are provided as Source data file.

**Combining ferroptosis induction with TRX pathway inhibition demonstrates broad therapeutic efficacy across SCLC NE subtypes in vivo and serves as prognostic marker set in human SCLC.** Although several efforts are underway to develop selective GPX4 inhibitors with improved bioavailability for clinical applications, these do not meet required pharmacokinetics yet[32,34]. Therefore, we made use of the fact that we found the clinically

advanced GCLC inhibitor BSO to induce lipid ROS-dependent ferroptosis in non-NE SCLC cells (Fig. 5e). Moreover, to equally repurpose a clinically applied inhibitor for the TRX pathway to facilitate rapid clinical translation for the treatment of SCLC, we used Auranofin in vivo. In order to mimic non-NE/NE intratumoral heterogeneity, we co-transplanted 50:50 mixed murine stickers and floaters containing 50% ASCL1− cells and 50%

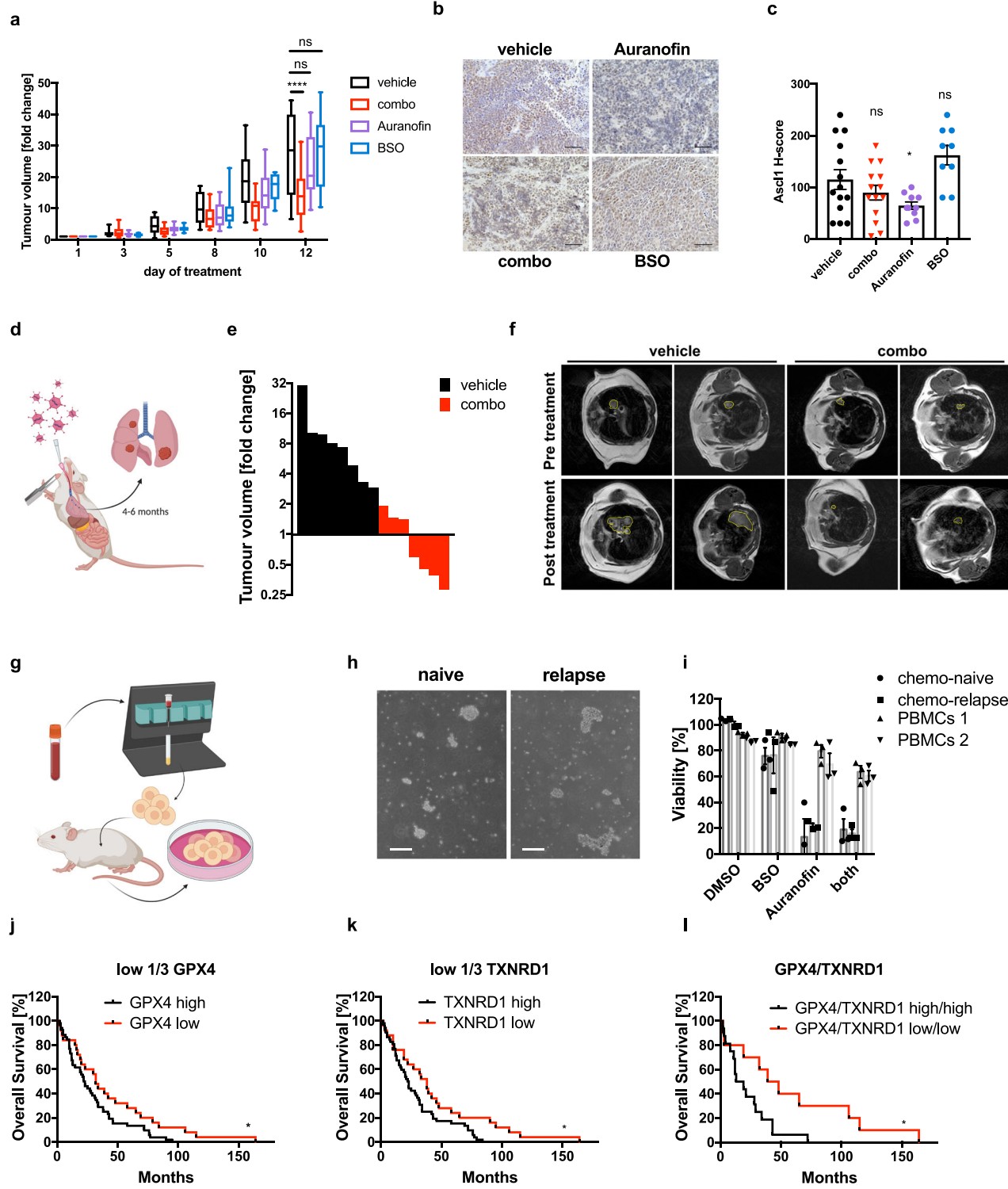

ASCL1+ cells (Supplementary Fig. 9a), respectively, into mice. Upon detection of palpable tumors, mice were randomized by tumor volume into four treatment groups receiving either vehicle, BSO, Auranofin or the combination (combo) for two consecutive weeks. Strikingly, only the combination achieved a significant anti-tumor response (Fig. 6a). Of note, while both single treatments also presented with a trend to slow early tumor growth, this trend was neutralized in later measurements suggesting NE/non/NE plasticity under single-arm treatment to compensate for SCLC tumor growth in vivo. Indeed, when staining residual

tumors after the end of treatment, vehicle and combo-treated tumors retained comparable ASCL1 histology scores in bigger or smaller tumors, respectively (Fig. 6b). While BSO-treated tumors only showed a trend toward an enrichment of ASCL1+ cells, Auranofin-treated tumors presented with a significantly reduced ASCL1 score (Fig. 6c). Therefore, only combined treatment can efficiently overcome non-NE/NE plasticity under single pathway treatment in heterogeneous SCLC resulting in a significant anti-tumor effect. These data suggested that SCLC may be a particularly sensitive entity for this combined strategy due to its NE

**Fig. 6 Combined ferroptosis induction and TRX pathway inhibition demonstrates broad anti-tumor activity across SCLC NE subtypes in vivo and serves as prognostic marker set in human SCLC. a** Eight-weeks-old male nude mice were injected with 50:50 mixed 181.5 stickers and floaters at a total of $1.5 \times 10^6$ cells into both flanks. Once palpable, tumors were treated either with vehicle ($n = 14$), Auranofin [2.5 mg/kg] ($n = 10$) 3× per week i.p., BSO [5 mM] ($n = 10$) in the drinking water or the combination (combo) ($n = 14$) for 2 consecutive weeks. Fold change of initial tumor size is shown. Boxplot center line, mean; box limits, upper and lower quartile; whiskers min. to max. **b** Sections from paraffin-embedded tumors stained for ASCL1. Representative images are shown, scale bar = 100 μm. **c** ASCL1 H-score was quantified, n (vehicle) = 14, n (BSO) = 9, n (Aura) = 10, n (combo) = 14. **d** Eight to twelve-week-old RP mice were inhaled intratracheally with $2.5 \times 10^7$ plaque-forming units (PFU) Adeno-Cre virus to initiate SCLC development. **e** Tumor-bearing RP-mice were treated either with vehicle ($n = 8$) or combined BSO [5 mM] in the drinking water and Auranofin 3× per week i.p. [2.5 mg/kg] ($n = 7$) for 2 consecutive weeks. Fold change in tumor volumes was determined by quantifying initial tumor volume from MRI scans as compared to tumor volume at the end of the treatment cycle using Horos software. **f** Representative MRI images pre and post treatment of mice as in (**e**). **g** Isolation scheme of human CDXs. **h** Cellular morphology of human CDXs, scale bar = 400 μm. **i** Two CDXs or two healthy donor PBMCs were treated with DMSO, BSO [PBMCs, 500 μM; CDXs, 50 μM], Auranofin [250 nM] or BSO [PBMCs, 500 μM; CDXs, 50 μM]/Auranofin [250 nM] for 24 h, cell viability was quantified by Cell Titer Blue (CDXs) or flow cytometric quantification of propidium iodide (PI)-negative cells (PBMCs). **j** Kaplan–Meier survival curves for SCLC patients ($n = 77$)[7] containing low (low 1/3 $n = 25$, median survival 33 months) or high (high 2/3 $n = 52$, median survival 22.5 months) expression of GPX4 mRNA. **k** As in (**j**) expression of TXNRD1 mRNA was correlated using the same cut-off (low = 1/3, median survival 38 months; high 2/3, median survival 22.5 months). **l** Kaplan–Meier survival curves for SCLC patients with combined low or high GPX4 and TXNRD1 mRNA expression (low/low $n = 10$, median survival 43.5 months; high/high $n = 16$, median survival 16 months). Data are means ± SEM were applicable (**i**). Schemes were drawn with fully licensed Biorender.com. Two-tailed unpaired t tests (**a**, **c**), or two-sided log-rank (Mantel–Cox) test (**j-i**), ***$p < 0.001$, *$p < 0.05$, ns $p > 0.05$. Source Data are provided as Source data file.

subtypes being addicted to mutually exclusive anti-oxidant pathways. Therefore, we next validated the efficacy of the identified combined treatment regime also in representative human NE and non-NE SCLC xenografts. Indeed, we obtained a very clear and significant response also in established human NE and non-NE SCLC xenograft tumors in vivo (Supplementary Fig. 9b–e). Importantly, the lipid ROS generation byproduct malondialdehyde (MDA), recently validated as a specific in vivo marker for ferroptosis induction[66], was increased in combo-treated tumors of both subtypes indicating in vivo induction of ferroptosis by combo treatment (Supplementary Fig. 9f).

To next validate efficacy in an immune-competent autochthonous mouse model, we made use of an established genetically engineered mouse model (GEMM) recapitulating all relevant features of human SCLC[53]. In these mice, SCLC develops as a consequence of Rb1 and Tp53 co-deletion (RP-mice) upon adenoviral Cre inhalation within 9 months (Fig. 6d). Moreover, intratumoral heterogeneity is observed in these mice as tumors contain both ASCL1$^{high}$ and $^{low}$ cells[3]. Once tumors had a detectable mean volume of 15–30 mm$^3$, determined by magnetic resonance imaging (MRI), mice were randomized and treated with combo for 2 weeks (Supplementary Fig. 9g). Whereas all vehicle control mice progressed, 4 out of 7 tumors receiving combo significantly regressed within 2 weeks (Fig. 6e, f).

Besides the use of GEMMs, circulating tumor cells (CTCs) and CTC-derived xenotransplants (CDXs) from SCLC have proven to be a powerful tool for faithful recapitulation of patient response to chemotherapy[67,68]. In order to test a potential response to combo in this human model system, we obtained CDXs from a treatment-naïve and a post-chemotherapy relapse SCLC patient (Fig. 6g, h). Both CDXs, which were confirmed to express NE markers, were sensitive to Auranofin but not ferroptosis induced by BSO whereas normal peripheral blood monocytes (PBMCs) were resistant suggesting low levels of toxicity (Fig. 6i). Importantly, when treating these cells with combo, similar levels of killing could be achieved in chemotherapy-naïve and relapse CDXs (Fig. 6i) indicating that prior chemotherapy and relapse does not impact response to combo. To lastly determine whether protection from ferroptosis via GPX4 or protection from ROS-dependent cell death via TrxR1 (TXNRD1) is of prognostic value in SCLC, overall survival of patients who had undergone surgical resection[7] was analyzed. Interestingly, low GPX4 or low TrxR1 (TXNRD1) expression, both independently correlated with improved overall survival (Fig. 6j, k). Importantly, when

analyzing SCLC patients with combined low expression of GPX4 and TXNRD1 mimicking combo treatment, we obtained a group of patients with a drastically improved median survival time of 43.5 months as compared to 16 months median survival in the high/high group (Fig. 6l). This expression pattern did not correlate with treatments these patients received post-surgery (chemotherapy or radiation)[7]. Moreover, to remove survival bias originating from the advanced tumor stage at diagnosis, diagnosed stage IV patients were excluded from the analysis and stage III patients were equally present in both groups ($n = 4$ in high/high; $n = 3$ in low/low). Importantly, survival in lung adenocarcinoma (LUAD) did not significantly segregate by GPX4, TXNRD1 or combined GPX4 and TXNRD1 expression (Supplementary Fig. 9h–j) suggesting a function for redox pathway plasticity in determining patient outcome to be a unique feature of SCLC. Therefore, concomitant low expression of GPX4 and TXNRD1 serves as an independent and specific prognostic marker set for overall survival in SCLC.

In conclusion, we report that treatment-naïve SCLC exhibits signs of selection against extrinsic apoptosis and necroptosis and upregulates xCT for ferroptosis protection. While we identify the non-NE-A SCLC subtype to be exquisitely ferroptosis sensitive and present with high ACSL4 expression and a ferroptosis-prone lipidome, we find the NE-A SCLC subset to be resistant to ferroptosis but selectively vulnerable to TRX pathway inhibition. Thereby, we identify that SCLC non-NE/NE-subtypes mechanistically segregate by ferroptosis sensitivity or resistance. Importantly, heterogeneous SCLC cultures selectively deplete non-NE or NE subpopulations upon single pathway targeting and demonstrate plasticity under treatment (Fig. 7). Due to this particular biology of SCLC, combining ferroptosis induction with TRX pathway inhibition demonstrates high therapeutic efficacy in SCLC xenografts, GEMMs, patient-derived CDXs and identifies a unique SCLC patient subset with drastically improved prognosis. These data propose that combined ferroptosis induction/TRX pathway inhibition may specifically tackle the problem of intratumoral NE/non-NE heterogeneity and plasticity in SCLC.

## Discussion

Small cell lung cancer (SCLC) is one of the most aggressive types of cancer causing more than 20 k fatalities per year in the US alone. Although many SCLC patients initially respond to standard-of-care chemotherapy, they invariably develop

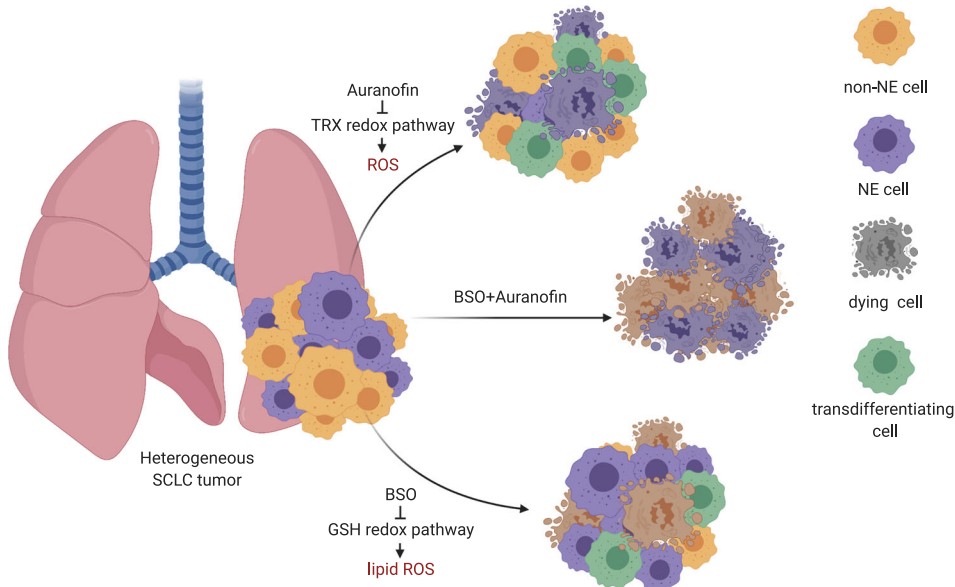

**Fig. 7 SCLC non-NE/NE subtypes segregate by lipid ROS/ROS vulnerability.** Heterogeneous SCLC tumors contain NE cells expressing high levels of ASCL1 and low levels of REST1 and non-NE cells with an inverse expression pattern. Upon induction of ferroptosis by BSO, non-NE cells die due to excessive lipid ROS accumulation. NE cells, in turn, undergo ROS-dependent cell death upon TRX pathway inhibition. Surviving cellular fractions in both single treatment arms can transdifferentiate under treatment, thereby escaping therapy. Dual targeting prevents selection and plasticity in SCLC. NE neuroendocrine, ROS reactive oxygen species, GSH glutathione, TRX thioredoxin. The scheme was drawn using fully licensed Biorender.com.

relapse[69]. Therapeutically, SCLC has been treated as a single tumor entity, although well-defined molecular subtypes have been described. While several subtype-specific therapeutic approaches have been identified for SCLC[36–41] SCLC frequently presents with intratumoral heterogeneity i.e., several molecular subtypes can co-exist within a single tumor further complicating treatment. Recent evidence suggests that SCLC subtype intratumoral heterogeneity evolves as a consequence of subtype plasticity[5]. These findings highlighted the need to devise therapies targeting SCLC subtype plasticity directly rather than only isolated molecular subtypes. Here, through systematic characterization of regulated cell death pathway availability in treatment-naïve SCLC, we identify the non-NE subtype to be exquisitely ferroptosis sensitive while we find the NE subtype to be selectively vulnerable to TRX pathway inhibition. Importantly, we find non-NE/NE subtype plasticity to be driven by compensation between both pathways exposing their combined targeting as a plasticity-directed vulnerability.

We found NE-differentiation to determine ferroptosis resistance in SCLC. NE to non-NE fate switch in SCLC is a particularly interesting feature as it is closely linked with resistance to conventional chemotherapy. Non-NE cells within tumors can share a common origin with NE cells proposing a high degree of plasticity within SCLC[70]. Importantly, these non-NE cells are characterized by a mesenchymal phenotype and expression profile. Of note, a mesenchymal expression profile has been identified as a ferroptosis response signature[71]. Moreover, remodeling of the plasma membrane during epithelial-to-mesenchymal-transition (EMT) leads to an increase in biosynthesis of PUFAs which are the main target of lipid peroxidation, the fatal event in the execution of ferroptosis[72]. In accordance with these data, we identify non-NE SCLC to be exquisitely ferroptosis sensitive and to express EMT signature genes, such as *vimentin*. Key enzymes generating the PUFA lipid target pool for lipid peroxidation are ACSL4 and LPCAT3 which generate AA-CoA for lysophospholipid acylation, respectively. Demonstrating their significance in ferroptosis, both ACSL4 and LPCAT3 have previously been shown to be indispensable for ferroptosis[24–26,73]. Importantly, we

find both enzymes to be expressed at higher levels in non-NE SCLC. Yet, we also found that the majority of PUFA-containing DAG phospholipids were not elevated in non-NE SCLC while PUFA-containing ether lipids along with expression of ether lipid synthesis enzymes was selectively enriched in non-NE SCLC. These data, together with the finding that suppression of ether lipid synthesis rendered non-NE cells ferroptosis resistant, suggest that a pro ferroptotic function of ACSL4 and LPCAT3 in non-NE SCLC may stem from incorporation of AA-CoA into ether lysolipids. In support of this notion, LPCAT3 has been shown to be able to use lyso-ether lipids as substrates for acylation by AA-CoA[74], thereby promoting phospholipid recovery in a basic process termed Lands cycle[75]. Based on these and our data we propose that upregulated ether lipid synthesis in non-NE SCLC feeds into the ferroptosis vulnerable membrane lipid pool generated and re-generated by ACSL4 and LPCAT3 creating a "ferroptosis-prone" membrane lipid pool through the addition of ether-linked PUFAs.

Interestingly, Notch homolog 1 (NOTCH1) signaling has repeatedly been shown to promote expression of an EMT signature in various cancers[76]. In SCLC, NOTCH signaling promotes a slow growing and chemotherapy resistant non-NE cell fate via activation of REST1. Since activation of cMyc expression via CRISPRa as well as cMyc[T58A] expression was sufficient to sensitize SCLC cells to ferroptosis in our experiments, it is tempting to speculate whether NOTCH pathway activation by cMyc might be responsible for this. Therefore, it is tempting to speculate whether NOTCH pathway activation and a resulting EMT may set these intrinsically chemotherapy resistant cells up to be ferroptosis responsive. Recently, it was shown that cMyc[T58A] expression drives temporal evolution of SCLC towards a non-NE YAP1+ state[5]. Suggesting increasing ferroptosis sensitivity to be proportional to this evolutionary trajectory, additional expression of constitutively active YAP1 5SA further sensitized non-NE SCLC cells in our experiments.

Interestingly, the mevalonate and lipid synthesis pathway was recently shown to be controlled by MEK5/ERK5 and required for optimal growth in SCLC[77]. Of note, the mevalonate pathway is

equally responsible for the synthesis of coenzyme Q10 (CoQ10), the reduced form of which (ubiquinol) is generated by FSP1 and was recently shown to act as a radical trapping agent preventing ferroptosis[55,56]. Given our finding that non-NE SCLC presents with elevated ACSL4 expression and a ferroptosis-prone lipidome, it is tempting to speculate whether targeting of MEK5/ERK5 or the mevalonate pathway induces a partially ferroptotic cell death in non-NE SCLC.

Besides its role in driving NE cell fate in SCLC, ASCL1 expression has been instrumental in neuronal reprogramming. Here, upon ASCL1 expression cells have to overcome a period of enhanced sensitivity to ferroptosis to fully transdifferentiate into neurons[78]. Thereby, to tolerate ASCL1 expression these cells must acquire ferroptosis resistance through selection and adaption rather than through direct ASCL1-mediated ferroptosis resistance. We propose that this may equally be the case in SCLC as acute ASCL1 overexpression did not render cells resistant to ferroptosis, yet, all ferroptosis resistant SCLC cells express ASCL1. In line with an idea of increased ferroptosis selective pressure arising as a result of ASCL1 expression, ASCL1 suppressed GCLC resulting in a drop in cellular GSH levels. We propose that this reduced cellular redox potential imposed by ASCL1 expression forces NE SCLC into a selective dependency on the TRX anti-oxidant pathway. However, we find that even the TRX pathway is further compromised in NE SCLC through the upregulated expression of TXNIP pushing NE SCLC toward the "edge of the cliff" of anti-oxidant defense failure. Therefore, we suggest that while non-NE SCLC can compensate for TRX pathway inhibition, most NE SCLC cells have exhausted all means of anti-oxidant defense and die due to excess accumulation of cellular ROS. Thereby, we find selective addiction to the TRX pathway to be a mechanistic feature of SCLC "neuroendocrineness" and it will be interesting to determine whether this principle extends to other neuroendocrine cancers. The fact that we find both SCLC subtypes investigated to be vulnerable to two distinct types of anti-oxidant defense, suggests that SCLC as a whole might experience elevated oxidative damage through excessive generation of ROS species. One source of this ROS may stem from increased rates of mitochondrial respiration, a process recently shown to promote ferroptosis[79]. Noteworthy, elevated expression of cMYC has been demonstrated to increase cellular oxidative phosphorylation (OXPHOS) and glycolytic metabolism, thereby contributing to elevated ROS production[80]. While MYC amplification is relatively rare in NSCLC (about 5%), MYC amplifications have been observed in 20% of SCLC cases[81]. MYC expression causes differentiation of a non-NE phenotype in mouse models of SCLC[36] offering an explanation as to why non-NE SCLC experiences oxidative stress from ROS. Yet, we find NE SCLC to also depend on anti-oxidant defense via the TRX pathway suggesting elevated ROS to also be a problem in this subtype. Interestingly, during neuronal differentiation, mitochondrial genes along with OXPHOS were shown to be upregulated[82]. Although NE differentiation in SCLC is not identical to this process, several transcriptional programs, including those initiated by ASCL1, are shared. Thereby, both, non-NE and NE SCLC subtypes are characterized by transcriptional programs known to elevate ROS.

Interestingly, we also find that in non-NE as well as in NE SCLC subpopulations can transdifferentiate under single redox pathway targeting and thereby escape treatment whereas dual targeting prevents this plasticity. While dual targeting of the TRX and GSH synthesis pathway has previously been identified as a synergistic killing strategy against several cancer entities, so far, reasons as to what the nature of the synergy may be and patient groups which may actually benefit from this regime in clinical trials have not been identified. Our data propose that while part of the synergy may stem from compensatory upregulation of the

other redox pathway in cancers without plasticity, another level of efficacy of dual targeting in SCLC may actually stem from preventing treatment escape by transdifferentiation driven by elevated ROS. Thereby, a proportion of the synergy might be caused by drifting subpopulation dynamics rather than intracellular compensation mechanism. Our data from isogenic non-NE/NE cells suggest that SCLC may undergo a spontaneous switch from a GSH-dependent non-NE state toward usage of the TRX antioxidant pathway upon ASCL1-mediated suppression of GSH synthesis. This SCLC-specific feature of NE-non-NE plasticity may explain why, so far, the in vivo combination of BSO with Auranofin and Carboplatin has only achieved very limited efficacy in lung adenocarcinoma, which is not characterized by NE/non-NE plasticity[66]. Supporting this notion, combined low expression of GPX4 and TXNRD1 was prognostic for drastically improved overall survival of SCLC but not LUAD patients.

Despite recent advances in the treatment of patients with SCLC, survival rates remain very poor. Our work indicates that non-NE SCLC is exquisitely sensitive to therapeutic induction of ferroptosis whilst NE SCLC demonstrates selective addiction to the TRX redox pathway and plasticity between both subtypes relies on switching between both pathways. We anticipate that providing a novel mechanistic separation for SCLC subtype biology and plasticity may on the one hand benefit patient stratification and on the other hand facilitate informed treatment approaches that take selective cell death pathway availability in SCLC subtype heterogeneity and plasticity into consideration. Due to the inherent quality of lipid ROS and ROS-dependent types of cells death being molecularly independent of caspase-dependent types of cell death, we propose that this therapeutic route should remain amenable for both chemotherapy-naïve and -relapsed SCLC.

## Methods

**Cell lines and culture conditions**. Human SCLC cell lines (H82, COR-L88, H524, H889, L303, H526, H2171, H1836, H1092) were grown in suspension in RPMI (GIBCO) and were obtained from ATCC (https://www.lgcstandards-atcc.org/?geo_country=de). Murine SCLC cell lines (RP252.7, RP214, RP181.5, RP285.5, RP251, RP250) were previously derived from lung tumors of a genetically engineered mouse model for SCLC driven by loss of Trp53 and Rb1 by the lab of H. Christian Reinhardt. These cells were grown in RPMI. MEFs were kindly provided by Manolis Pasparakis and kept in DMEM (GIBCO), human NSCLC cell lines NCI-H460 and NCI-H727 were kindly provided by Julian Downward and kept in in DMEM (GIBCO). GPX4 KO SCLC cells were kept in RPMI with 5 μM Ferrostatin-1 (Cayman chemicals) during their generation and for part of the experiments. CDXs and PBMCs were grown in RPMI medium. All media were supplemented with 10% fetal calf serum (FCS, Sigma-Aldrich) and all cells were kept at 37 °C with 5% CO$_2$. All cell lines were tested for mycoplasma at regular intervals (mycoplasma barcodes, Eurofins Genomics) and human SCLC have been validated by STR profiling provided by Eurofins Genomics. Sticker and floater isogenic cell lines were isolated freshly from the RP mouse model. Whole lung tumors were obtained from mouse lungs, washed twice in ice-cold PBS, and cut into small pieces, followed by 20 min digestion with 10x trypsin (ThermoFisher) at 37°C. Cells were continuously cultured in RPMI medium and passaged until they formed a uniform cell line. None of the cell lines used fall under the category of commonly misidentified cell lines.

**Reagents**. Isoleucine zipper- (iz) TRAIL was kindly provided by Henning Walczak and TNF was kindly provided by Manolis Pasparakis. The small molecules were obtained from the respective company in brackets: zVad (Enzo), Birinapant (Bertin Pharma), erastin (Biomol), RSL3 (Selleckchem), ML210 (Sigma), Ferrostatin-1 (Sigma), DFO (TargetMol), Necrostatin-1 (Abcam), NAC (BIOTREND Chemicals AG), Auranofin (Cayman Chemicals), BSO (Sigma), PX-12 (Hölzel) and D9 (Sigma), Puromycin (Sigma), Dharmafect I (Dharamcon), Doxycycline (VWR), DRAQ7 (Biolegend), Polybrene (Merck), CaCl$_2$ (Sigma). Cisplatin and etoposide were obtained from Merck. BODIPY C11, H2DCFDA, MCB, and Calcein-AM were purchased from Invitrogen, Sigma, and Abcam, respectively. Lipid standards (made by Avanti Polar lipids) were purchased from Sigma. ON-TARGETplus Smartpool siRNAs mouse Agpat2 siRNAs: CAGCCAGGUUCUACGCCAA, GGGUACACGCAACGACAAU, GCUUUGAGGUCAGCGGACA, CCGUGGA UAACAUGAGCAU; Agpat3 siRNAs: CUUCGUGGUGAGCGGGUUA, CCA GAUGGGAAGACCGCAU, CUGAUGCGGUAUCCAAUUA, GAUAGGAGU

GACUGAGAUA; Gclc siRNAs: GCGAUGAGGUGGAAUACAU, UAACAGA
CUUUGAGAACUC, UGGCAGACAAUGAGAUUUA, CCAUCUCCAUUUAU
AGAAA; Txnrd2 siRNAs: GACAAAGGCGGGAAGGCGA, GGGAUGCAUCA
CAGUGCUA, GACUGGUUGCUGAGGCUAA, GGGAAAUCCUCAACCUUAA;
Txnrd1 siRNAs: GCAUCAAGUUUAUAAGACA, GCGAUAUAUUGGAGGAU
AA, CUAAGGAGCAGCCCAAUA, GGACAGCACAAUUGGAAUC) were
obtained from Dharmacon, mouse Slc7a11 siRNA was obtained from Qiagen using
the following two sequences: CAACGUUGAUGAUGGACUAT and
GAUUUAUCUUCGAUACAAATT

**Antibodies**. Anti-Gpx4 (Abcam, ab41787, 1:2000), Anti-xCT (Abcam, ab37185,
1:2000), Anti-Cas9 (Cell signaling, 14697, 1:1000,), Anti-ß-Actin (Sigma, A1978,
1:10,000), Anti-GCLC (Santa Cruz, sc-166345, 1:1000), Anti-Ascl1 (BD Phar-
mingen, 556604, 1:1000), Anti-Txnrd1 (Cell signaling, 15140S, 1:1000), Anti-
Txnrd2 (Cell signaling, 12029, 1:1000), Anti-Acsl4 (Santa Cruz Biotechnology, sc-
271800, 1:2000), Anti-Txn1 (Cell signaling, 2429S, 1:1000), Anti-GAPDH (Cell
signaling, 97166S, 1:2000), Anti-FSP1 (previously described[55], kindly provided by
M. Conrad, undiluted hybridoma supernatant), Anti-REST1 (ThermoFisher, BS-
2590R, 1:1000), Anti-REST1 (Abcam, ab21635, 1:1000), Anti-TXNIP (Cell sig-
naling, 14715S, 1:1000), Anti-CD71 (Santa Cruz, sc-65882, 1:2000), Anti-cMyc
(Abcam, ab32072, 1:2000), Anti- NCAM (Invitrogen, PA5-79717, 1:1000), Anti-
Vimentin (Abcam, ab137321, 1:1000), Anti-YAP1 (Cell signaling, #4912, 1:1000),
Anti-Synatophysin (Invitrogen, MA5-14532, 1:1000), Anti-AGPAT2 (Thermo
Fisher, PA5-76010, 1:2000), Anti-AGPAT3 (Thermo Fisher, PA5-101343, 1:2000).
HRP-conjugated secondary antibodies: goat-anti-mouse-HRP (Linaris GmBH,
20400-1 mg, 1:10,000), goat-anti-rabbit-HRP (Linaris GmBH, 20402-1 mg,
1:10,000), goat-anti-rat-HRP (Sigma, A9037-1 ml, 1:10,000).

**Cell viability and cell death assays**. Cell viability was determined by Cell Titer
Blue assay (Promega) following the manufacturer's instructions. For this assay,
cells were plated at 10,000 or 5000 cells/96-well in 100 µl media. Alternatively, cell
viability was determined by Cell Titer-Glo (CTG) assay (Promega) plating 5000
cells/96-well in 100 µl media, treatment of cells the next for 48 h followed by Cell
Titer-Glo (CTG) assay (Promega) according to the manufacturer's instructions.
Cell death was quantified as cells positive for propidium iodide (PI) uptake
(1 µg/ml). For this, cells were plated at 50,000 cells in 500 µl media per 24-well.
Quantification was done by flow cytometry using an LSR-FACS Fortessa (BD
Bioscience) counting 5000 cells per sample.

**Generation of CRISPR/Cas9-mediated GPX4 KO cells**. RP252.7 and RP285.5
were stably transfected with an expression plasmid for Cas9 (#52962 supplier),
followed by selection in Blasticidin (10 µg/µl) for 5 days. Next, cells were infected
with lentivirus carrying either empty vector (lentiGuide-Puro (#52963) or vector
containing one of three different GPX4-targeting gRNAs (for guide generation two
primers/guide with specific overhangs were annealed, see Supplementary Table 1)
and cells were kept in 5 µM Ferrostatin-1 from this time onwards:

    GPX4 gRNA 1: GACGATGCACACGAAACCCC
    GPX4 gRNA 2: ACGATGCACACGAAACCCCT
    GPX4 gRNA 3: CGTGTGCATCGTCACCAACG
    After selection with Puromycin (7 µg/ml) for 4 days whole cell populations were
validated for KO via Western blot and used in Ferrostatin-1 withdrawal
experiments.

**Generation of inducible Ascl1 overexpressing cells**. For stable transduction of
cells, viral particles were produced in HEK293T cells. First, HEK 293T cells were
plated one day prior in 10 cm cell culture dish at a confluence of 70–80% at the day
of transfection. Lentiviral particles were packaged using the packaging plasmids of
third-generation packaging system: pCMV-VSV-G (#8454) by Bob Weinberg, and
pRSV-Rev (#12253) and pMDLg/pRRE (#12251) by Didier Trono. As transfer
plasmid containing the ASCL1 cDNA, pCW-Cas9 (#50661) from Eric Lander &
David Sabatini, was used. The Cas9 fragment was firstly replaced with ASCL1
cDNA. For transfection of a 10 cm dish, 10 µg of the transfer plasmid and 5 µg of
each lentiviral packaging plasmid were prepared together with 400 µl 250 mM
CaCl2. For the formation of calcium-phosphate-DNA co-precipitate, the 400 µl of
2× HEBS buffer was added drop by drop to the CaCl2–DNA mixture under
constant vortex. The mixture was then added drop-wise to the cells. After 6 h
transfection, cell culture medium was replaced with RPMI with 20% FCS and 1%
P/S. The following three days virus-containing supernatant was harvested and
filtered with 0.45 µm sterile syringe filter. Target cells for transduction were plated
at a confluence of 30% and virus-containing supernatant with 6 µg/ml polybrene
was added on three consecutive days.

**Generation of YAP1 5SA overexpressing cells**. Retroviral particles were pro-
duced in HEK Phoenix cells. First, HEK Phoenix cells were plated one day prior in
10 cm cell culture dish at a confluence of 70–80% at the day of transfection. For
transfection of a 10 cm dish, 10 µg of pBABE empty vector or pBABE YAP 5SA-
YFP (kindly provided by Dr. Erik Sahai) were prepared in 400 µl 250 mM CaCl2.
For the formation of calcium-phosphate-DNA co-precipitate, the 400 µl of 2×
HEBS buffer was added drop by drop to the CaCl2–DNA mixture under constant

vortex. The mixture was then added drop-wise to the cells. After 8 h transfection,
cell culture medium was replaced with RPMI with 20% FCS and 1% P/S. The
following three days virus-containing supernatant was harvested and filtered with
0.45 µm sterile syringe filter. RP181.7 cells were plated at a confluence of 30% and
virus-containing supernatant with 6 µg/ml polybrene was added on three con-
secutive days. Cells were selected using 2.5 µg/ml puromycin.

**Clonogenic survival assay**. Murine SCLC cells were plated at 2,500 cells/24-well
in 500 µl medium a day in advance. The next day they were treated with DMSO,
RSL3 or erastin with or without Ferrostatin-1 (5 µM) for 24 h after which wells
were washed with PBS and replenished with fresh media for incubation for another
6 days. On day 7, cells were washed with PBS, fixed and stained for 30 minutes
using crystal violet solution (0.05% (w/v) crystal violet, 1% Formaldehyde, 1%
methanol in PBS).

**Western blotting**. Cells were washed, lysed in IP-lysis buffer (30 mM Tris-HCl,
120 mM NaCl, 2 mM EDTA, 2 mM KCl, 1% Triton-X-100, pH 7.4, protease and
phosphatase inhibitor (Roche)) and frozen at −20 °C. After re-thawing, lysate
concentrations were adjusted to equal protein concentrations using the bicincho-
ninic acid (BCA) protein assay (Biorad). Equal amounts of protein were mixed to a
final concentration of 1× reducing sample buffer (Invitrogen) containing 200 mM
DTT. Samples were heated to 80 °C for 10 min, separated via gel electrophoresis
and transferred to PVDF membranes using the TurboBlotting system (Biorad).
Membranes were blocked in PBS with 0.1% Tween 20 (PBST) with 5% (w/v) dried
milk powder for at least 30 min. Next, membranes were incubated overnight at 4 °C
with primary antibodies in PBST with 5% bovine serum albumin (BSA). After
washing with PBST, membranes were incubated with horseradish peroxidase
(HRP)-coupled secondary antibodies for at least 1 h. After another washing step,
membranes were developed using chemiluminescent substrate Immobilon Lumi-
nata Classico (Millipore) and X-ray films CL-XPosure™ (Thermo Scientific).

**Lipid ROS quantification by BODIPY C11 staining**. For lipid ROS quantification,
50,000 cells were seeded in 500 µl media in 24-well plates. The next day, cells were
treated with DMSO, 100 nM RSL3 or 100 nM RSL3/5 µM Fer-1 for 5 h. During the
last 30 min of incubation, BODIPY C11 was added at 5 µM to each well. Cells were
then washed, detached, and fold increased green fluorescence over baseline (stained
but untreated) was determined by flow cytometry (excitation 520 nm) counting
5000 cells per sample.

**General ROS quantification by H2DCFDF staining**. To stain cells, cells were
treated as for BODIPY C11 staining but instead H2DCFDF was added to wells
during the last 30 min treatment incubation at 20 µM/well. Cells were then washed,
detached and fold increased green fluorescence over baseline (stained but
untreated) was determined by flow cytometry (excitation 496 nm) counting 5000
cells per sample.

**Labile iron quantification by Calcein-AM staining**. For quantification of labile
iron in cells, cells were seeded one day in advance, the next day were washed three
times with PBS to remove residual FCS containing free iron. Cells were incubated
for 20 min with 500 nM Calcein-AM in PBS. Cells were detached and washed, and
fold increased green fluorescence over baseline (stained but untreated) was
determined by flow cytometry (excitation 496 nm) counting 5,000 cells per sample.

**Cellular GSH quantification**. To determine relative levels of GSH in responders
and non-responders, the thiol-reactive dye monochlorobimane (MCB) was added
to cells seeded a day in advance (50,000 cells per well in 24-well plates) at a
concentration of 50 µM for 30 min. Cells were then washed, detached and analyzed
by flow cytometry (405 nm excitation) counting 5000 cells per sample. To quantify
concentrations of cellular GSH and GSSG, the GSH/GSSG Glo Assay (Promega)
was used according to the manufacturer's instructions using 75,000 (H82 and
H2171) or 50,000 (RP) cells.

**siRNA transfections**. For all knockdowns described, 200 µL Opti-MEM and 1.5 µl
Dharmafect Reagent I were mixed per single 6-well and incubated for 10 min at
room temperature. 2.2 µL siRNA (stock 20 mM) per 200 µL Opti-MEM mixture
were added and incubated for 30 min at room temperature. 200 µL of the mixture
were added to each well of a 6-well plate and cells were plated on top in 1 mL
media. Knockdowns were incubated for 48–72 h, as indicated.

**ASCL1-staining for flow cytometry**. RP181.5 stickers or floaters ($1 \times 10^5$ cells)
were seeded in 12-well plates a day in advance and then treated with 5 mM BSO or
1 µM Auranofin for 24 h. Cells were harvested, washed with PBS and stained for
live/dead cells using the viability dye eFluor780 (eBioscience), 1:1000, for 30 min,
at 4 °C. Cells were then washed twice with FACS buffer (PBS, 2% FCS) and cell
pellets were resuspended in 200 µl Fixation/Permeabilization buffer (eBioscience)
(overnight incubation at 4 °C). The next day, cells were washed with 1× Permea-
bilization buffer (eBioscience) and incubated for 15 min in FACS buffer before

adding the primary MASH-1 antibody (BD Biosciences, 556604), 1:250, for 30 min at 4 °C in 1× Permeabilization buffer. After washing cells twice with 1× Permeabilization buffer, pellets were resuspended in the secondary antibody (Cy3, Jackson Laboratories) 1:500 for 30 min at 4 °C in 1× Permeabilization buffer. Cells were again washed twice with 1× Permeabilization buffer and resuspended in FACS buffer. Measurements were acquired using a BD LSR Fortessa flow cytometer and data were analyzed with the FlowJo software. For sorting experiments cells were resuspended in PBS with 2% FCS and 25 mM HEPES and passed through a BD FACS sorter (Influx).

**Redox shift assays.** $1.5 \times 10^6$ cells (H82, RP181.5 floaters) and $1 \times 10^6$ cells (H2171, RP181.5 stickers) were seeded in 6-well plates and treated for 0.5 or 6 h with 1 µM Auranofin. Cells were then harvested, washed in PBS and lysed in 1 ml 8% (w/v) trichloroacetic acid (TCA) on ice and frozen at −20 °C. Samples were thawed and centrifuged for 15 min at 20,000 × g at 4 °C. TCA supernatant was removed, pellets were centrifuged for 5 min at 20,000 × g at 4 °C and residual TCA was removed. 20 µl of alkylation buffer (6 M urea, 0.2 M Tris-HCl, pH 7.5, 0.2 M EDTA, 2% SDS, bromophenol blue) was added and samples were sonicated for 10 cycles at an amplitude of 60% to dissolve pellets (UP50H, Hielscher). For minimum and maximum samples, tris(2-carboxyethyl)phosphine (TCEP) was added (10 µM final concentration) followed by incubation for 10 min at 50 °C. For the minimum shift sample, N-ethyl maleimide (NEM) was added (15 mM final concentration) and to all other samples methyl-polyethylenglycol-maleimide (mmPEG24, 15 mM final concentration) was added. Samples were incubated for 1 h in the dark at room temperature, then 20 µl of 2× Laemmli buffer was added. Samples were loaded on tris-tricine gels and run overnight at constant 130 V at 4 °C.

**Quantification of oxidized glycerophospholipids via lipidomics.** Levels of oxidized phosphatidylcholine (PC) and phosphatidylethanolamine (PE) species were determined by liquid chromatography coupled to electrospray ionization tandem mass spectrometry (LC-ESI-MS/MS) using a procedure described in Doll et al.[55] with several modifications: $2.5 \times 10^6$ cells were resuspended in 250 µl of an ice-cold 2:5 (v/v) mixture of 100 µM diethylenetriaminepentaacetic acid (DTPA) in PBS, pH 7.4, and 40 µM butylated hydroxytoluene (BHT) in methanol. To 100 µl of the cell suspension another 3.4 ml of the above-mentioned PBS/methanol mixture, 1.25 ml of ice-cold chloroform and internal standards (10 pmol 1,2-dimyristoyl-sn-glycero-3-phosphocholine (DMPC) and 10 pmol 1,2-dimyristoyl-sn-glycero-3-phosphoethanolamine (DMPE) were added. The samples were vortexed for 1 min and incubated at −20 °C for 15 min. After adding 1.25 ml of chloroform and 1.25 ml of water, the mixture was vortexed vigorously for 30 s and then centrifuged (4000 × g, 5 min, 4 °C) to separate layers. The lower (organic) phase was transferred to a new tube and dried under a stream of nitrogen. The residues were resolved in 150 µl of methanol and transferred to autoinjector vials. LC-MS/MS analysis was performed by injecting 20 µl of sample onto a Core-Shell Kinetex C18 column (150 mm × 2.1 mm ID, 2.6 µm particle size, 100 Å pore size, Phenomenex) at 30 °C and with detection using a QTRAP 6500 triple quadrupole/linear ion trap mass spectrometer (SCIEX). The LC (Nexera X2 UHPLC System, Shimadzu) was operated at a flow rate of 200 µl/min. The mobile phase system, gradient program and source- and compound-dependent parameters of the mass spectrometer were set as previously described[55]. Oxidized PC and PE species and the internal standards were monitored in the negative ion mode using their specific multiple reaction monitoring (MRM) transitions[55]. The LC chromatogram peaks were integrated using the MultiQuant 3.0.2 software (SCIEX). Oxidized PC and PE species were quantified by normalizing their peak areas to those of the internal standards. The normalized peak areas were related to the mean values of the floaters + DMSO cell samples.

In the MRM analyses, specific precursor ions—in our case the molecular ions ([M–H]⁻) of the different PE species—were selected in the first quadrupole (Q1). These PE species were fragmented in the second quadrupole (Q2), which serves as collision cell. Simultaneously, the third quadrupole (Q3) was set to let pass only a characteristic PE fragment ion. The advantage of MRM analyses is that only ions, which fit both, the selected precursor and the characteristic fragment ions, can reach the detector whereas all other (lipid) species in the LC eluate are ignored. To enable the selective detection of only those PE species, which are acylated by oxidized fatty acids, and the differentiation of these oxidized from non-oxidized PE species (including plasmalogens), we chose the oxidized fatty acyl chains as characteristic fragment ions in Q3.

**Quantification of glycerophospholipid via lipidomics.** Glycerophospholipids (PC and PE, including ether-linked species) in cells were analyzed by Nano-Electrospray Ionization tandem mass spectrometry (Nano-ESI-MS/MS) with direct infusion of the lipid extract (Shotgun Lipidomics): approximately $3 \times 10^6$ cells were homogenized in 300 µl of Milli-Q water using the Precellys 24 Homogenisator (Peqlab) at 6.500 rpm for 30 s. The protein content of the homogenate was routinely determined using bicinchoninic acid. To 30 µl of the homogenate 470 µl of Milli-Q water, 1.875 ml of methanol/chloroform 2:1 (v/v) and internal standards (187 pmol PC 17:0-20:4 and 198 pmol PE 17:0-20:4, Avanti Polar Lipids) were added. Lipid extraction and Nano-ESI-MS/MS analysis were performed as previously described[83]. Endogenous glycerophospolipids were quantified by referring

their peak areas to those of the internal standards. The calculated glycerophospolipid amounts were normalized to the protein content of the cell homogenate.

**Quantitative PCR.** For isolation of total RNA from cells the NucleoSpin RNA kit (740955.5, Macherey-Nagel) was used according to the manufacturer's protocol. The isolated RNA was reverse transcribed into cDNA using the LunaScript RT SuperMix Kit (E3010L, NEB) following the protocol provided by the manufacturer. For quantitative PCR the Power SYBR Green PCR Master Mix (4368702, Thermo Fisher) was used. To this end, 5 µl Power SYBR Green PCR Master Mix was mixed with 2 µl of nuclease-free water (NEB). After adding 1 µl (10 µM) of forward and reverse primers (Supplementary Table 1) and 2 µl of cDNA (5 µg/µl) real-time qPCR was performed in triplicates on the Quant Studio5 qRT PCR machine. Results were normalized to the expression of the house-keeping gene GAPDH, which served as a reference control.

**IncuCyte cell imaging.** Cells were plated in 96-well plates ($1 \times 10^4$ stickers, $2 \times 10^4$ floaters) and stimulated with or without Ferrostatin-1 [5 µM] 2 h prior to adding either DMSO, BSO [10 mM], Auranofin [1 µM] or RSL3 [1 µM]. Dead cells were stained by adding 100 nM DRAQ7 (Thermofisher) to all wells. Cells were imaged for 24 h every 5 h and 3 images per well were captured.

**ASCL1-staining for immunofluorescence microscopy.** RP181.5 stickers (5000 cells) were seeded in 24-well plates. The following day, they were treated with 500 nM Auranofin and 500 µM BSO for 96 h. For staining cells were washed with PBS followed by incubation in 200 µl Fixation/Permeabilization buffer (eBioscience) overnight at 4 °C. The next day, cells were washed with 1× Permeabilization buffer (eBioscience) and incubated for 15 min in FACS buffer (PBS, 2% FCS) before adding the primary MASH-1 antibody (BD Biosciences, 556604), 1:250, for 30 min at 4 °C in 1× Permeabilization buffer. After washing cells twice with 1× Permeabilization buffer, cells were incubated in the secondary antibody (Cy3, Jackson Laboratories) 1:500 for 30 min at 4 °C in 1× Permeabilization buffer. Cells were again washed twice with 1× Permeabilization buffer and subsequently incubated in PBS for microscopy. Microscopy was performed using a Leica DMI 6000B microscope (×20 objective) and images were analyzed with the ImageJ software.

**Tumor xenograft studies.** $1.5 \times 10^6$ cells (either H82 or H2171) in 200 µl PBS were injected into both flanks of 8-week old male NMRI-Foxn1 nu/nu mice (Janvier). Mice were enrolled either into vehicle or combination treatment groups once tumors reached a minimum size of $2 \times 2$ mm. For two consecutive weeks, mice were injected either with vehicle (PBS with 2% DMSO, 8.5% ethanol and 5% polyethylene glycol 400) and received normal drinking water or were injected with Auranofin (2.5 mg/kg) three times a week and received BSO (5 mM) in the drinking water ad libitum. Tumor size was tracked by caliper measurements and volume was calculated as (length * width * width)/2. Mice were sacrificed at the end of the treatment. Due to variance in tumor size at the start of the experiment, fold change of tumor volume upon treatment is plotted. Mice were housed in individually ventilated cages (IVCs) at 12 h/12 h light/dark cycle, 55 ± 10% humidity and 22 ± 2 °C ambient temperature and received autoclaved food, water, and bedding. All animal experiments were approved by the local authorities (LANUV, North-Rhine-Westphalia, Germany) and performed under license number 81-02.04.2017.A477. All people involved in animal experiments received prior training and have passed the additionally required personal licensing course (FELASA-B). All animal experiments were conducted in compliance with international and institutional ethical guidelines on animal welfare and measures to minimize animal suffering.

**Sticker/Floater in vivo studies.** 50:50 mixed $7.5 \times 10^5$ and $7.5 \times 10^5$ RP181.5 stickers and floaters, respectively were injected into both flanks of 8-week old male NMRI-Foxn1 nu/nu mice (Janvier) in 200 µl PBS. Mice were enrolled into either vehicle, BSO, Auranofin or combination treatment groups once tumors reached a minimum size of $2 \times 2$ mm. For two consecutive weeks, mice were injected either with vehicle (PBS with 2% DMSO, 8.5% ethanol and 5% polyethylene glycol 400), received BSO (5 mM) in the drinking water ad libitum, were i.p. injected with Auranofin (2.5 mg/kg) three times a week or received combined BSO/Auranofin. Tumor size was tracked by caliper measurements and volume was calculated as (length * width * width)/2. Mice were sacrificed at the end of the treatment. Due to variance in tumor size at the start of the experiment, fold change of tumor volume upon treatment is plotted. Mice were housed in individually ventilated cages (IVCs) at 12 h/12 h light/dark cycle, 55 ± 10% humidity and 22 ± 2 °C ambient temperature and received autoclaved food, water and bedding. All animal experiments were approved by the local authorities (LANUV, North-Rhine-Westphalia, Germany) and performed under license number 81-02.04.2017.A477. All people involved in animal experiments received prior training and have passed the additionally required personal licensing course (FELASA-B). All animal experiments were conducted in compliance with international and institutional ethical guidelines on animal welfare and measures to minimize animal suffering.

**Immunohistochemistry**. For immunohistochemistry tissues were fixed in 4% formaldehyde solution (Merck) and subsequently embedded in paraffin. After sectioning tissue (4 μm) it was deparaffinized and rehydrated according to standard procedure. For antigen retrieval samples were heated in a citrate-based buffer for anti-MDA stainings (Vector Laboratories, H-3300) to 100 °C for 30 min or in a TE-based buffer [10 mM Trizma Base, 1 mM EDTA, pH = 9] in a pressure cooker for anti-ASCL1 stainings. Subsequently, samples were blocked for endogenous peroxidases (BLOXALL endogenous peroxidase and alkaline phosphatase blocking solution, Vector Laboratories, SP-6000, 15 min) and for unspecific binding, for anti-MDA stainings (5% BSA, 5% NGS in PBST, 1 h) and Avidin/Biotin (Avidin/Biotin Blocking Kit, Vector Laboratories, SP-2001, 15 min) and for anti-ASCL1 stainings using mouse-on-mouse blocking buffer (Abcam) 30 min. Samples were incubated overnight at 4 °C with anti-MDA antibody (Abcam, ab6463) 1:250 in blocking buffer (PBS, 1% BSA, 0.003% NaN3, 0.05% Tween20) or using anti-ASCL1 (BD Pharmingen, 556604) 1:300 in TBS 1% BSA. The following day samples were washed three times in PBS-T and incubated with secondary antibody (Perkin Elmer, NEF813) 1:1,000 in blocking buffer for 1 h (anti-MDA stainings) or HRP polymer detector for 15 min. for anti-ASCL1 (Abcam, mouse on mouse kit) and washed as before. For staining anti-MDA, samples were incubated with PBS + 1/60 Biotin + 1/60 Avidin (VECTASTAIN® Elite® ABC HRP Kit, Vector Laboratories, PK-6100) for 30 min. Both stainings were developed using DAB chromogen (Abcam, ab6423) according to the manufacturer's instructions and counterstained using Hematoxylin.

**ASCL1 histology scoring**. ASCL1 stainings were manually quantified by H-score on a scale of 0–300 as described[84]. In brief, the H-score was calculated from % of positive cells multiplied by intensity score of 0–3. To this end, tumors were first categorized into intensities followed by an estimation of % of positive cells.

**RP-mouse model, MRI scans, treatment**. The well-established genetically engineered mouse model recapitulating SCLC harbors a Rb1flox/flox allele in which exons 18 and 19 are flanked by loxP sites as well as a Trp53flox/flox allele in which exons 2–10 are flanked by loxP sites (RP model). RP mice used in this experiments were kept on a mixed C57Bl6/J;Sv129 background. For induction of lung tumor formation, 8–12-weeks-old mice were anesthetized with Ketavet (100 mg/kg) and Rompun (20 mg/kg) by intraperitoneal injection followed by intratracheal inhalation of replication-deficient adenovirus expressing Cre (Ad5-CMV-Cre, 2.5 × 10^7 PFU, University of Iowa) to their lungs[53]. Five months after tumor induction, tumor formation was monitored bi-weekly by magnetic resonance imaging (MRI) (A 3.0 T Philips Achieva clinical MRI (Philips Best, the Netherlands) in combination with a dedicated mouse solenoid coil (Philips Hamburg, Germany), were used for imaging. T2-weighted MR images were acquired in the axial plane using turbo-spin echo (TSE) sequence [repetition time (TR) = 3819 ms, echo time (TE) = 60 ms, field of view (FOV) = 40 × 40 × 20 mm³, reconstructed voxel size = 0.13 × 0.13 × 1.0 mm³, number of average =1) under isoflurane (2.5%) anesthesia. MR images (DICOM files) were analyzed by determining and calculating region of interests (ROIs) using Horos software. Once tumors reached a mean volume of 15–30 mm³, mice were randomized into two groups and treated with either vehicle or the combination of BSO (5 mM, via drinking water) and Auranofin (2.5 mg/kg, 3× per week, i.p.) for 2 weeks. Mice were housed in individually ventilated cages (IVCs) at 12 h/12 h light/dark cycle, 55 ± 10% humidity and 22 ± 2 °C ambient temperature. All animal experiments were approved by the local authorities (LANUV, North-Rhine-Westphalia, Germany) and performed under license number 81-02.04.2017.A477. All people involved in animal experiments received prior training and have passed the additionally required personal licensing course (FELASA B). All animal experiments were conducted in compliance with international and institutional ethical guidelines on animal welfare and measures to minimize animal suffering.

**Generation of human SCLC CDXs**. Circulating tumor cells (CTCs) were isolated from blood of two patients diagnosed with SCLC following previously described protocols[67]. In brief, both patients from which CTCs were isolated were diagnosed with stage IV SCLC; both had received prior treatment with carboplatin + etoposide, samples were received and CTCs enriched at the time of relapse in one case (relapse) and samples received and CTCs enriched at the time of first diagnosis in the other case (naïve). CTCs were isolated and enriched by growth in immune-compromised mice (strain NSG). After the development of a tumor in immuno-compromised NSG mice, cells were dissociated, expanded and maintained in cell culture in HITES media. At least 90% of cells were confirmed to be tumor cells with neuroendocrine marker expression (chromogranin A, CD56 and SYP) as determined by RNA analysis and immunohistochemistry (IHC). The use of patient material was approved by the ethics committee of the University Hospital Cologne following written informed consent. We have complied with all relevant ethical regulations pertaining to the use of human patient material.

**PBMC isolation**. PBMCs were isolated from buffy coats from two healthy donors under approval obtained from the ethics committee of the University Hospital Cologne (#01-090) following written informed consent. We have complied with all relevant ethical regulations pertaining to the use of human patient material.

**Human SCLC RNA-seq data ethical approval**. All SCLC and normal lung patient datasets and human SCLC cell line data used in this study have been previously published and, as such, are appropriately referenced, have previously obtained the appropriate ethics approvals and are available online under https://doi.org/10.1038/nature14664 for George et al.[7] (SCLC data), https://doi.org/10.1038/ng.2405 for Rudin et al.[85] (normal lung data) and under https://doi.org/10.1016/j.cell.2016.12.005 for Mollaoglu et al.[36] (human SCLC cell line RNA-seq data). Raw sequencing data were analyzed with TRUP[86] and gene expression was quantified as FPKM.

**Analysis software and bioinformatic analysis**. Heatmaps visualizing cell death pathway component expression were generated using RStudio version 1.1.456 and gplots and RColorBrewer packages. A ranked list of fold differential expression was generated for human cell line RNA-seq data using GSEA Desktop v3.0 (https://doi.org/10.1073/pnas.0506580102 and https://doi.org/10.1038/ng1180). FACS data were analyzed and quantified using FlowJo 10.4.2. Cell Titer Blue viability assays were analyzed using Excel. MRI scans were quantified using Horos v3.3.5. Lipidomics measurements were analyzed by MultiQuant 3.0.2 software (SCIEX). Figures were assembled and data plotted and analyzed using GraphPad Prism 7 for Mac OS X.

**Statistics and reproducibility**. All data are presented as mean ± SEM of at least three independent biological replicates unless stated otherwise. Thereby, means are calculated and plotted from at least three means each of which was calculated from at least two technical replicates from at least three independent experiments. No technical or biological replicates were excluded, biological replicates gave comparable results. Statistical tests used are indicated in the respective figure legends. For all tests used the following $p$ value cut-offs were applied: ****$p < 0.0001$, ***$p < 0.001$, **$p < 0.01$, *$p < 0.05$, ns $p > 0.05$. Representative western blots are shown out of at least three independent biological replicates. All statistical analysis was performed using GraphPad Prism 7 for Mac OS X.

**Reporting summary**. Further information on research design is available in the Nature Research Reporting Summary linked to this article.

## Data availability

All data supporting the findings in this study are available from the corresponding author upon reasonable request. The previously published human SCLC patient[7], normal human lung[85], and human SCLC cell line[36] RNA-seq datasets used in this study are available at the European Genome-phenome Archive, which is hosted by the European Bioinformatics Institute (EBI), under accession code EGAS00001000925 (SCLC patients), EGAS00001000334 (normal lung) and EGAS00001002115 (human SCLC cell lines). Source data are provided with this paper.

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

## Acknowledgements

We thank Karin Schlegelmilch for constructive discussions and advice on YAP1, Steve Hooper for cloning and Erik Sahai for providing the pBabe YAP1 5SA-YFP plasmid, all members of the Department of Translational Genomics for constructive feedback, Henning Walczak and Manolis Pasparakis for providing recombinant TRAIL and TNF, respectively, Johannes Kühle for providing buffy coats and Trudy Oliver for advice. This work was supported by an Erasmus Fellowship by the European Union (E.S.T.), a Max-Eder-Junior Research Group grant (701125509, S.v.K.) by the German Cancer Aid, a project grant (432038712, S.v.K.), a collaborative research center grant on SCLC (SFB1399, A01, H.C.R., A05, S.v.K., B01, R.K.T., B02, J.G., C02, H.C.R., A02, M.L.S.), a collaborative research center grant on cell death (SFB1403, A05, S.v.K.) and a medical student fellowship (L.S.) under Germany's Excellence Strategy—(CECAD, EXC 2030—390661388), all funded by the German Research Foundation (Deutsche Forschungsgesellschaft, DFG), by an eMed consortium grant (01ZX1901A, S.v.K.) funded by the German state (BMBF) and a project grant (A06, S.v.K.) funded by the center for molecular medicine (CMMC), Cologne, Germany. Work in the M.L.S. lab is further supported by the Thyssen Foundation (10.19.2.025MN) and the German Cancer Aid (70112888).

## Author contributions

C.M.B. performed many in vitro and all in vivo experiments, designed experiments, supervised Z.C. and L.S. and created the figure schemes, E.S.T. performed many of the in vitro experiments and bioinformatic analysis, Z.C. and J.S. performed knockdown experiments and experimental repeats and J.S. assisted with in vivo experiments, A.S. assisted with RP mouse in vivo experiments, A.A. performed redox pathway plasticity FACS experiments, M.N.H. performed redox shift assays, L.S. assisted with tissue stainings, M.A.D. performed human SCLC cross titrations, C.P.A. cultured primary CDXs, A.K. performed human cell line RNA-seq data analysis, H.L.T. generated RPM cell lines from murine tumors, F.P. assisted with human SCLC cross titrations, F.B. performed TCGA LUAD survival analysis, M.L.S. designed human SCLC cross titration experiments, J.R. designed redox shift experiments, J.G. provided CDXs and gene expression data, S.B. performed lipidomics measurements, R.K.T. and H.C.R. designed experiments and S.v.K. conceived the project, designed experiments, supervised work, acquired funding, and wrote the manuscript. All authors read and edited the manuscript.

## Funding

## Competing interests

H.C.R. received consulting fees from Abbvie, AstraZeneca, Vertex and Merck and research funding from Gilead. R.K.T. is a co-founder of and was a consultant for NEO New Oncology, now part of Siemens Healthcare. R.K.T. and M.L.S. are co-founders of and consultants for PearlRiver Bio GmbH and M.L.S. receives funding from PearlRiver Bio GmbH. H.L.T. is a consultant for PearlRiver Bio GmbH. R.K.T. is a co-founder and consultant of Epiphanes Inc. R.K.T. is a stockholder of Roche, AstraZeneca, GSK, Merck, Qiagen, Novartis, Bayer and Johnson & Johnson. All other authors declare no competing interests.
