## [Peer Review File · Nature Communications]

REVIEWER COMMENTS

Reviewer #1 (Remarks to the Author):

The manuscript by Bebber and coworkers investigates potential therapies for small cell lung cancer. This manuscript has conducted transcriptomic analysis to determine that non-neuroendocrine SCLC, nonNE, are susceptible to induced cell death by inhibiting the glutathione pathway, which would increase oxidation of membrane lipids. The increase in lipid hydroperoxides promotes necrosis by ferroptosis. Neuroendocrine SCLC does not have inhibition of the glutathione pathway but relies on the thioredoxin, TRX, pathway to protect against reactive oxygen species. Inhibition of TRX results in cell death. This manuscript uses multiple SCLC cell lines and inhibitors to demonstrate this phenomenon. They also use isogenic cells in which they shift the phenotype between nonNE and NE cell types to demonstrate the sensitivity of nonNE cells to ferroptosis and NE to TRX inhibition. Finally, this manuscript tests this phenomenon out in a xenograft mouse model. All in all this is a well designed manuscript in which the data clearly supports the hypothesis proposed. This manuscript introduces a novel approach to treat SCLC by identifying the predominant type and then either triggering ferroptosis or inhibiting TRX or a combination of inhibitors. Enthusiasm would have been raised by adding these inhibitors in the mouse model for SCLC or using Patient derived xenografts to test these out not on cell lines in a xenograft model but in a tumor model in immunocompromised mice. However this is a strong manuscript whose findings may have direct impact on human lung cancer. .

Reviewer #2 (Remarks to the Author):

In this manuscript, the authors attempt to identify regulated cell death (RCD) pathways in SCLC, which might be important to guide SCLC therapy and drug development. They found that SCLC is resistant to apoptosis and necroptosis, but is sensitive to ferroptosis, a form of RCD caused by iron-dependent lipid peroxidation. Interestingly, their results appear to indicate that only the non-neuroendocrine (NE) SCLC but not NE SCLC is sensitive to ferroptosis. Instead, they found that NE SCLC gains dependency on TRX anti-oxidant pathway. They found that combining ferroptosis induction with inhibition of the TRX pathway kills established non-NE and NE tumors, in both xenograft and genetically engineered mouse models of SCLC. Overall, their findings are interesting and clinically relevant. Several issues need to be satisfactorily addressed to further support and justify their conclusions, as detailed below:

1. Mechanistically, the authors need to determine how non-NE SCLC and NE SCLC have different sensitivity to ferroptosis. In discussion, they suggested that non-NE SCLC presents with elevated ACSL4 expression and a ferroptosis-prone lipidome, which might be the reason of sensitivity to ferroptosis. However, these data are only from the so-called "floaters" and "stickers" of one cell line (Fig 2L and 2M), which is not a strong evidence. The authors at least need to measure the expression of some common ferroptosis regulators (e.g. GPX4, ACSL4, SLC7A11, FSP1, etc) in a panel of non-NE SCLC and NE SCLC cell lines. To have more clinical relevance, in the data set of treatment-naïve SCLC samples (Fig 1), can authors cluster them into NE and non-NE subtypes and compare the expression of some common ferroptosis regulators? More fundamentally, if they identify certain ferroptosis-relevant molecular difference between NE SCLC and non-NE SCLC, what is the mechanism underlying such difference?
2. The authors used the "floaters" and "stickers" system as auto subtype transition model to address the ferroptosis sensitivity of NE and non-NE subtypes. However, it is well known that ferroptosis sensitivity is also regulated by cell-cell contact and cell density. Therefore, it is important to determine whether the factors involved in cell-cell contact, such as Hippo/YAP pathway as reported previously, all contribute to the ferroptosis sensitivity of "stickers" and "floaters".
3. Related to point-2, an experimentally more convincing and elegant system for dissecting NE SCLC and non-NE SCLC is an inducible system (which can avoid the effects of cell morphology change, etc.), compared to the sticker-floater analysis. In Fig S6, the authors have set up an inducible ASCL1 expression system. It is important to use this system to address ferroptosis and TRX inhibition sensitivity as well. Similarly, in the NE SCLC cells, the authors need to check whether knock down of ASCL1 will sensitize the cells to ferroptosis.

4. An important conclusion of the study is that NE SCLC use the anti-oxidant TRX pathway to resist to ferroptosis induction (the anti-ferroptotic function of TRX pathway has been reported previously), due to defect of glutathione (GSH) synthesis. This mechanism suggest that NE but not non-NE cells will be resistant to BSO, but both cell types should still be equally sensitive to erastin (cystine depletion, thus depriving GSH synthesis in both cell types) and RSL3 (direct inhibition of GPX4, making GSH irrelevant to GPX4 activity). However, this is not what the authors have observed (they claim NE are more resistant to all three treatments) - why?
5. For the xenograft expt in Fig 5A and 5B, single treatment of BSO or Auranofin should be included, which is very important to address the different sensitivity of NE and non-NE SCLC to ferroptosis as well as TRX inhibition in vivo. Moreover, NE and non-NE markers should be stained in the tumor sections, which are important to show whether lineage transition is induced by only targeting one of the two pathways.
6. As the authors propose a crucial role of the TRX anti-oxidant pathway in SCLC ferroptosis regulation, a thorough discussion of the source of oxidant production, especially in a SCLC-specific manner, will be very helpful. Would it be a certain metabolic pathway, or mitochondrial metabolism in general, or certain ferroptosis/redox-related enzymes? And might these events be particularly relevant in SCLC (the authors did argue that SCLC and NSCLC are different in terms of ferroptosis response).

Reviewer #3 (Remarks to the Author):

Comments:

In this manuscript Bebbber et al., identify how heterogeneous nature of SCLC may contributes to its resistance and possibly the higher relapse rate and show high survival efficiency by using a combination strategy targeting cell death pathways (ferroptosis and ROS induced) in two distinct cell subtypes (NE/Non-NE) present in SCLC.

The authors use different cancer lines (human and murine) and the conclusions are complemented well with some experiments in mice. Although not all the parts of the MS are new but overall, this fits well with the scope of Nature Communications, however I have some comments as below.

1. Overexpression of xCT is already known in multiple human cancers, and suggests that SCLC also upregulates metabolic pathways to handle the ROS. To support the results presented (Fig. 1L), the authors should perform the experiment under xCT knock-down/knock-out conditions in SCLC cell lines used. In addition, the expression of known regulators of ferroptosis (like xCT, GXP4, FSP1 and ACSL4) will be helpful for both NSCLC and SCLC in Fig 1L. It may be present as supplementary information.
2. In order to ascertain the contribution of 'free iron' in Ferroptosis of SCLC's, the authors should also look into the levels of iron regulating proteins like transferrin receptor1, ferritin etc, or measure the levels of free-iron in these cells. Just using deferoxamine (DFO) isn't enough.
3. Why is there huge variation in the cell death in controls- DMSO treated samples, even the same cell type in two different experiments have significantly different dead cells (compare Fig. 2 B & C with D & E)? Same is the case in Figs. S2 A; S4 E & F; S7 E & G. This should be fixed.
4. The authors should also show the total ROS for Fig. 3C and Fig. 3J.
5. Is the difference in REST1 between stickers and floaters really significant?
6. Fig. 3L, the authors mentioned they tested various proteins involved in ferroptosis pathway but show only ACSL4, other proteins should also be tested- GPX4, FSP1, LPCAT, Tfr 1, xCT
7. For Fig. 4F, in case of untreated, if possible select a different field/spot of cells.
8. Does BSO treatment also induce lipid ROS in fessoptosis sensitive SCLC?
9. The combination treatment of BSO/Auranofin seems to work well in SCLC (Fig. S7E,F), however, H82 cells which are sensitive to ferroptosis respond to BSO treatment and show a slight synergistic

increase with Auranofin. On the other hand, H2171 cells resistant to ferroptosis, should be sensitive to Auranofin treatment alone but it doesn't show such behavior. The combination works very well but again from the data it is not clear what type of cell death is contributing to H2171 death. Inhibitors (Fer.1 and NAC) used are known to protect ferroptosis. Same thing is observed in Fig. S7C, Ferrostatin 1 alone doesn't rescue RP 250.1 cells but does so in combination with NAC,

10. Ferrostatin 1 normally works at low concentrations (0.1 -0.4 μM), but the authors are using 5 μM , why?

11. In the lipidomics analysis, how did the authors differentiate between Plasmalogen + 1(O) and normal PE.

Point-by-point-response

Reviewer #1 (Remarks to the Author):

The manuscript by Bebbler and coworkers investigates potential therapies for small cell lung cancer. This manuscript has conducted transcriptomic analysis to determine that non neuroendocrine SCLC, nonNE, are susceptible to induced cell death by inhibiting the glutathione pathway, which would increase oxidation of membrane lipids. The increase in lipid hydroperoxides promotes necrosis by ferroptosis. Neuroendocrine SCLC does not have inhibition of the glutathione pathway but relies on the thioredoxin, TRX, pathway to protect against reactive oxygen species. Inhibition of TRX results in cell death. This manuscript uses multiple SCLC cell lines and inhibitors to demonstrate this phenomena. They also use isogenic cells in which they shift the phenotype between nonNE and NE cell types to demonstrate the sensitivity of nonNE cells to ferroptosis and NE to TRX inhibition. Finally, this manuscript tests this phenomena out in a xenograph mouse model. All in all this is a well designed manuscript in which the data clearly supports the hypothesis proposed. This manuscript introduces a novel approach to treat SCLC by identifying the predominant type and then either triggering ferroptosis or inhibiting TRX or a combination of inhibitors. Enthusiasm would have been raised by adding these inhibitors in the mouse model for SCLC or using Patient derived xenographs to test these out not on cell lines in a xenograph model but in a tumor model in immunocompromised mice. However this is a strong manuscript whose findings may have direct impact on human lung cancer.

We want to thank this reviewer for acknowledging the interest and novelty in our study and appreciate the constructive points raised. In the revised version of the manuscript we have highlighted the treatment validation in patient-derived xenograft cells ex vivo and the therapeutic testing of the combination in the widely used genetically engineered mouse model (GEMM) for SCLC in which SCLC is induced by adenoviral Cre inhalation leading to concomitant deletion of Rb1 and p53. In addition, we have now also included another mouse model, in which we have modelled heterogeneous non-NE/NE SCLC tumors by establishing tumors from 50:50 mixed isogenic non-NE/NE murine SCLC cells. Importantly, we have now included single treatment arms in this model to determine whether these single treatments may lead to selective subtype depletion in heterogeneous tumors in vivo. Indeed, combined treatments demonstrates a superior anti-tumor effect than single arm treatments. With these novel in vivo data included in new figure 6a-c, we believe to have comprehensively addressed this reviewer's suggestions.

Reviewer #2 (Remarks to the Author):

In this manuscript, the authors attempt to identify regulated cell death (RCD) pathways in SCLC, which might be important to guide SCLC therapy and drug development. They found that SCLC is resistant to apoptosis and necroptosis, but is sensitive to ferroptosis, a form of RCD caused by iron-dependent lipid peroxidation. Interestingly, their results appear to indicate that only the non-neuroendocrine (NE) SCLC but not NE SCLC is sensitive to ferroptosis. Instead, they found that NE SCLC gains dependency on TRX anti-oxidant

pathway. They found that combining ferroptosis induction with inhibition of the TRX pathway kills established non-NE and NE tumors, in both xenograft and genetically engineered mouse models of SCLC. Overall, their findings are interesting and clinically relevant. Several issues need to be satisfactorily addressed to further support and justify their conclusions, as detailed below:

1. Mechanistically, the authors need to determine how non-NE SCLC and NE SCLC have different sensitivity to ferroptosis. In discussion, they suggested that non-NE SCLC presents with elevated ACSL4 expression and a ferroptosis-prone lipidome, which might be the reason of sensitivity to ferroptosis. However, these data are only from the so-called “floaters” and “stickers” of one cell line (Fig 2L and 2M), which is not a strong evidence. The authors at least need to measure the expression of some common ferroptosis regulators (e.g. GPX4, ACSL4, SLC7A11, FSP1, etc) in a panel of non-NE SCLC and NE SCLC cells lines.

To have more clinical relevance, in the data set of treatment-naïve SCLC samples (Fig 1), can authors cluster them into NE and non-NE subtypes and compare the expression of some common ferroptosis regulators? More fundamentally, if they identify certain ferroptosis-relevant molecular difference between NE SCLC and non-NE SCLC, what is the mechanism underlying such difference?

We very much appreciate this reviewer’s interest in the study and the constructive criticisms raised. In order to provide additional information on ferroptosis pathway regulator expression we have performed RNA-sequencing on stickers and floaters and find that while expression patterns of other regulators (such as GPX4, xCT, FSP1, CD71) cannot explain elevated sensitivity in non-NE stickers, indeed ACSL4 and LPCAT3 expression was elevated in these cells (Supplementary Fig. 5k). In order to substantiate our finding in isogenic spontaneously transdifferentiating “stickers” and “floaters”, we have followed this reviewer’s advice and now generated a total of three isogenic pairs of mouse SCLC cell lines from murine tumors of three mice from the genetically engineered mouse model for SCLC (Fig. 3g,h). All three isogenic pairs undergo spontaneous non-NE/NE differentiation. Again, expression of ACSL4 and LPCAT3 was elevated in all three of these non-NE stickers, interestingly, along with enzymes promoting ether lipid synthesis, a recently described process promoting ferroptosis sensitivity (Zou et al. nature 2020) (Fig. 4f). Importantly, we find that indeed non-NE SCLC is more sensitive to ferroptosis due to elevated ether lipid synthesis and as such becomes more resistant to ferroptosis upon suppression of AGPATG3 and AGPAT2 (Fig. 4g, h) and presents with elevated levels of ether lipid species (Fig. 4d, e). Thereby, we have now identified ether lipid plasticity to contribute to differential ferroptosis sensitivity of non-NE and NE SCLC subtypes.

Following this reviewer’s suggestion, we have now clustered human SCLC patient data according to ASCL1 expression. Intriguingly, the small ASCL1-low group present in this patient cohort (n=10) also contained relatively higher levels of ACSL4 and GNPAT expression. While AGPAT3 expression was high in all of these 10 patients its expression was more heterogeneous in ASCL1+ patients. These data support our findings of a changed lipid metabolism in non-NE/NE SCLC. These data are now included in new supplementary figure 6.

2. The authors used the “floaters” and “stickers” system as auto subtype transition model to address the ferroptosis sensitivity of NE and non-NE subtypes. However, it is well known that

ferroptosis sensitivity is also regulated by cell-cell contact and cell density. Therefore, it is important to determine whether the factors involved in cell-cell contact, such as Hippo/YAP pathway as reported previously, all contribute to the ferroptosis sensitivity of "stickers" and "floaters".

This is a point well taken. To address this point, we have included a comparative analysis of murine SCLC cells derived from the RP mouse model as compared to cells derived from the RPM mouse model, the latter of which has been characterized to present with a variant more non-NE phenotype as compared to the RP mouse model (Mollaoglu et al.). Of note, RP and RPM cells all retain a sticker phenotype despite RPM cells showing slightly elevated non-NE marker expression (Fig. 3j). Importantly, RPM cells were even more sensitive to ferroptosis than RP cells despite both cell types being adherent (Fig. 3i). Vice versa, all cells within the human SCLC cell line panel tested grow in suspension yet segregate by NE status in terms of ferroptosis sensitivity/resistance (Fig. 3a). In addition, we have followed the suggestion of this reviewer and monitored YAP1 expression in stickers and floaters and indeed find that stickers express elevated levels of YAP1 which was expected given that YAP1 expression defines non-NE differentiation in SCLC (Fig. 3g). To functionally test YAP1 expression in SCLC ferroptosis sensitivity we have generated cells expressing constitutively active YAP1 5SA which indeed were slightly more sensitive to ferroptosis (Fig. 3k). Moreover, YAP 5SA overexpression also increased expression of the non-NE marker REST1. Therefore, we propose that YAP1 expression may influence ferroptosis sensitivity in SCLC by promoting non-NE differentiation. We want to thank the reviewer for bringing this up as addressing it has prompted us to test additional non-NE models.

3. Related to point-2, an experimentally more convincing and elegant system for dissecting NE SCLC and non-NE SCLC is an inducible system (which can avoid the effects of cell morphology change, etc.), compared to the sticker-floater analysis. In Fig S6, the authors have set up an inducible ASCL1 expression system. It is important to use this system to address ferroptosis and TRX inhibition sensitivity as well. Similarly, in the NE SCLC cells, the authors need to check whether knock down of ASCL1 will sensitize the cells to ferroptosis.

We agree that an inducible experimental system for NE transdifferentiation would be desirable. We have therefore performed these experiments as suggested yet, in agreement with a previous study (Lim et al. 2017 nature), ASCL1 inducible expression nor suppression alone was sufficient to affect NE transdifferentiation and a floater phenotype in murine SCLC cells in vitro (data not show). In line with a crucial function for full NE differentiation in ferroptosis resistance, ASCL1 induction did also not render cells resistant to ferroptosis or more sensitive to Auranofin (Supplementary Fig. 7e). ASCL1 expression therefore only partially recapitulates one step happening during NE transdifferentiating (downregulation of GSH synthesis thereby creating the need to compensate via the TRX system) but does not directly induce the full phenotype switch. Hence, ASCL1 can be considered a marker of ferroptosis resistance in SCLC but its expression is not sufficient to drive ferroptosis resistance but likely upstream other factors driving NE differentiation and ether lipid remodelling. These data clarifying the role of ASCL1 have been added in Supplementary Fig. 7e and a section clarifying this point has been included in the text

In order to include other means to study various levels of non-NE differentiation, we have made use of comparative analysis of murine RP versus RPM cells (see comment above) but also CRISPRa activated cMYC expression as a means to drive non-NE differentiation

(Supplementary Fig. 5h, i) as well as YAP1 expression to push for variant non-NE differentiation (Fig. 3k). In all of these cellular models ferroptosis sensitivity is increased by experimental cues of non-NE differentiation substantiating our initial observation in a sticker/floater line.

4. An important conclusion of the study is that NE SCLC use the anti-oxidant TRX pathway to resist to ferroptosis induction (the anti-ferroptotic function of TRX pathway has been reported previously), due to defect of glutathione (GSH) synthesis. This mechanism suggest that NE but not non-NE cells will be resistant to BSO, but both cell types should still be equally sensitive to erastin (cystine depletion, thus depriving GSH synthesis in both cell types) and RSL3 (direct inhibition of GPX4, making GSH irrelevant to GPX4 activity). However, this is not what the authors have observed (they claim NE are more resistant to all three treatments) - why?

As the reviewer rightly points out, we conclude that NE SCLC compensates for defective GSH synthesis by using the TRX pathway for anti-oxidant defense instead. The fact that NE SCLC was not only more sensitive to TRX pathway inhibition but at the same time also more resistant to ferroptosis was an equally surprising finding to us given that we expected low GSH levels to facilitate full depletion of GSH by erastin or BSO treatment at relatively lower concentrations. Moreover, due to low levels of GSH in NE SCLC we initially expected lower GPX4 activity which should be easier to inhibit fully using RSL3. Yet, this is not what we have observed in all cellular systems tested, including additional isogenic non-NE/NE read-outs now included in the revised version of the manuscript (Fig. 3h, Supplementary Fig. 5b).

As the reviewer rightly points out, these data pointed towards a generally lowered sensitivity to lipid peroxidation in NE SCLC rather than only a selective defect in GSH synthesis. Therefore, to understand ferroptosis resistance of NE SCLC, we have performed additional lipidomics measurements to quantify relative amounts of non-oxidized PUFA phospholipid species in stickers/versus floaters (see comment above). Strikingly, we found that floaters selectively showed decreased amounts of PUFA-containing ether lipids while diacyl-phospholipids were mostly comparable. To determine whether NE SCLC might have a defective ether lipid metabolism, we comparatively determined expression of enzymes involved in diacyl and ether linked PUFA synthesis. Interestingly, enzymes involved in ether-linked PUFA synthesis including AGPAT2, AGPAT3 and GNPAT but also ACSL4 and LPCAT3 were upregulated in all isogenic cellular models of non-NE ferroptosis sensitive SCLC. Importantly, silencing of AGPAT3 or AGPAT2 indeed render non-NE SCLC more resistant to ferroptosis. These data have been included in a whole new Figure 4 and Supplementary Fig. 5m, n.

Therefore, we propose that during NE transdifferentiation, SCLC must pass through a "ferroptosis checkpoint" (as observed in transdifferentiating neurons Gascon et al. 2016 Cell Stem Cell) wherein initial GSH synthesis defect will increase cellular ROS creating selective pressure to rewire lipid metabolism in a way to render cellular membranes less sensitive to lipid peroxidation. We propose that only NE cells having successfully navigated this checkpoint and downregulated ether lipid synthesis survive and undergo full NE transdifferentiation.

5. For the xenograft expt in Fig 5A and 5B, single treatment of BSO or Auranofin should be

included, which is very important to address the different sensitivity of NE and non-NE SCLC to ferroptosis as well as TRX inhibition in vivo. Moreover, NE and non-NE markers should be stained in the tumor sections, which are important to show whether lineage transition is induced by only targeting one of the two pathways.

We agree that this has been very important to address and have, instead of just adding single treatment arms in the non-heterogeneous xenograft model, included an additional in vivo model in which we have established heterogeneous non-NE/NE mouse SCLC tumors by mixed sticker/floater injection and included single and combined treatment arms in vivo in this new model. We find that indeed combined treatment most effectively suppresses SCLC tumor growth overall as compared to single treatment arms. While single treatment arms show a slight early effect, which is later lost, our new data including tissue stainings for the proportion of ASCL1+ cells suggest that selective outgrowth of the respective non-targeted SCLC subtype accounts for this loss of efficacy of single arm treatments and justifies the use of the combination. These data are now part of new Figure 6a-c.

6. As the authors propose a crucial role of the TRX anti-oxidant pathway in SCLC ferroptosis regulation, a thorough discussion of the source of oxidant production, especially in a SCLC-specific manner, will be very helpful. Would it be a certain metabolic pathway, or mitochondrial metabolism in general, or certain ferroptosis/redox-related enzymes? And might these events be particularly relevant in SCLC (the authors did argue that SCLC and NSCLC are different in terms of ferroptosis response).

We agree that this has been missing and have added the following passage to the discussion:

“The fact that we find both SCLC subtypes investigated be vulnerable to two distinct types of anti-oxidant defense, suggests that SCLC as a whole might experience elevated oxidative damage through excessive generation of ROS species. One source of this ROS may stem from increased rates of mitochondrial respiration. Noteworthy, elevated expression of cMYC has been demonstrated to increase cellular oxidative phosphorylation (OXPHOS) and glycolytic metabolism thereby contributing to elevated ROS production⁷⁸. While MYC amplification is relatively rare in NSCLC (about 5%), MYC amplifications have been observed in 20% of SCLC cases⁷⁹. MYC expression causes differentiation of a non-NE phenotype in mouse models of SCLC³⁶ offering an explanation as to why non-NE SCLC experiences oxidative stress from ROS. Yet, we find NE SCLC to also depend on anti-oxidant defense via the TRX pathway suggesting elevated ROS to also be a problem in this subtype. Interestingly, during neuronal differentiation, mitochondrial genes along with OXPHOS were shown to be upregulated⁸⁰. Although NE differentiation in SCLC is not identical to this process, several transcriptional programs including those initiated by ASCL1 are shared and we indeed observed elevated expression of respiratory chain genes in NE-SCLC cells (data not shown). Thereby, both, non-NE and NE SCLC subtypes are characterized by transcriptional programs known to elevate ROS.”

Reviewer #3 (Remarks to the Author):

Comments:

In this manuscript Bebbler et al., identify how heterogeneous nature of SCLC may contribute to its resistance and possibly the higher relapse rate and show high survival efficiency by

using a combination strategy targeting cell death pathways (ferroptosis and ROS induced) in two distinct cell subtypes (NE/Non-NE) present in SCLC.

The authors use different cancer lines (human and murine) and the conclusions are complemented well with some experiments in mice. Although not all the parts of the MS are new but overall, this fits well with the scope of Nature Communications, however I have some comments as below.

1. Overexpression of xCT is already known in multiple human cancers, and suggests that SCLC also upregulates metabolic pathways to handle the ROS. To support the results presented (Fig. 1L), the authors should perform the experiment under xCT knock-down/knock-out conditions in SCLC cell lines used. In addition, the expression of known regulators of ferroptosis (like xCT, GPX4, FSP1 and ACSL4) will be helpful for both NSCLC and SCLC in Fig 1L. It may be present as supplementary information.

We have now performed xCT knockdowns, yet found that xCT suppression alone triggers compensation via the TRX pathway and therefore is insufficient to kill SCLC cells (see Supplementary Fig. 8h, i). In order to validate the induction of ferroptosis genetically we have generated GPX4 knockout cells using three independent GPX4-targeting guide RNAs. These cells but not controls are only viable in the presence of ferrostatin-1 and induce lipid ROS and die upon withdrawal of ferrostatin-1 (see also Figure 2).

As suggested, we have also comparatively analyzed expression levels of known regulators of the ferroptosis pathway in SCLC as compared to NSCLC cells. In order to analyze comparable cellular models, we used murine cell lines derived in house from lung tumors of the widely studied LsL-KRASG12D; p53^{F1/FL} (KP) genetically engineered mouse model (GEMM) for NSCLC as compared to in house derived cell lines from the widely used GEMM for SCLC (RP model), both on a C57BL/6 strain background. While GPX4, xCT and ACSL4 levels are expressed at comparable levels in NSCLC and SCLC, interestingly, transferrin receptor (CD71) expression is elevated in all SCLC cell lines as compared to 2 out of three NSCLC lines. Moreover, expression of FSP1, recently demonstrated to render cells more resistant to ferroptosis (Doll et al. 2019, Bersuker et al. 2019) was elevated specifically in NSCLC cells. While specific mechanisms of differential ferroptosis sensitivity in NSCLC versus SCLC deserve more in-depth studies in future work and an in-depth investigation of this question is not within the scope of this study, these interesting findings suggest that increased iron import and low FSP1 expression in SCLC may uniquely prime SCLC for ferroptosis. We have included these new data in Supplementary Fig. 1c.

2. In order to ascertain the contribution of 'free iron' in Ferroptosis of SCLC's, the authors should also look into the levels of iron regulating proteins like transferrin receptor1, ferritin etc, or measure the levels of free-iron in these cells. Just using deferoxamine (DFO) isn't enough.

We thank the reviewer for this suggestion as addressing it revealed that SCLC cells indeed express elevated levels of CD71 (see comment above and Supplementary Fig. 1c). Moreover, we have now performed Calcein AM staining which has been used to measure free labile iron pools within cells and indeed see, that DFO treatment decreases fluorescent quenching of this dye (Supplementary Fig. 2c), indicating that indeed less iron is available upon DFO treatment which also blocks cell death in accordance with the first study describing

ferroptosis as an iron-dependent cell death type (Dixon et al. 2012). Most importantly, we establish sensitivity of SCLC to ferroptotic cell death by GPX4 knockout experiments and specific induction of lipid peroxidation.

3. Why is there huge variation in the cell death in controls- DMSO treated samples, even the same cell type in two different experiments have significantly different dead cells (compare Fig. 2 B & C with D & E)? Same is the case in Figs. S2 A; S4 E & F; S7 E & G. This should be fixed.

We agree that this was confusing in the manuscript so far given that the same cell line was used. However, the cell lines grow in suspension and therefore will retain significant amounts of dead cells in the culture after splitting- a fact we did not mention before. This varies depending on when the experiments were done, passage etc. and therefore, varies between panels as e.g. experiments for the DFO panels in Figure 2 were done at a much later time point/passage than the Fer-1 panels. Means from three independent experiments merged to produce one panel were usually performed within 1 week and therefore represent error bars between consecutive passages. We felt that the way we presented the data so far was the most transparent way of data presentation but are open to normalizing all the data in suspension cells to background cell death levels if this is preferred. We have for clarity also included a sentence in the manuscript explaining this point.

4. The authors should also show the total ROS for Fig. 3C and Fig. 3J.

Yes, this is an important point. We have now performed experiments measuring increase in general ROS in representative human and mouse responders and non-responders and found that in murine SCLC, total ROS is only increased in responders, in representative human SCLC cells total ROS is also increased in non-responders highlighting that lipid ROS accumulation is selectively impaired in non-responders. These data are now added to Supplementary Fig. 3e, f.

5. Is the difference in REST1 between stickers and floaters really significant?

In order to broaden the scope of this investigation and provide more significant data, we generated two additional isogenic cell line pairs from fresh RP mouse tumors and analyzed a total of three pairs of stickers and floaters for expression of NE (ASCL1, synaptophysin, NCAM) and non-NE markers (REST1, vimentin, YAP1). Taken together, we could confirm that stickers represent non-NE cells while floaters acquire NE marker expression. Indeed, differences in REST1 expression are less obvious than some of the other markers now included (Fig. 3g). Nevertheless, all three sticker/floater lines clearly differ in expression of non-NE and NE markers and thereby support our conclusion that stickers represent non-NE SCLC while floaters acquire NE differentiation.

6. Fig. 3L, the authors mentioned they tested various proteins involved in ferroptosis pathway but show only ACSL4, other proteins should also be tested- GPX4, FSP1, LPCAT, Tfr 1, xCT

We agree that this is of course important to show. Therefore, we have now performed RNA-sequencing on stickers and floaters and find that while expression patterns of GPX4, xCT, FSP1 and CD71 cannot explain elevated sensitivity in non-NE stickers, indeed ACSL4 and also LPCAT3 expression was elevated in these cells (Supplementary Fig. 5k).

Following up on this observation, we could validate elevated ACSL4 and LPCAT3 expression in non-NE cells in three newly generated isogenic cell line pairs undergoing spontaneous non-NE/NE differentiation. Interestingly, other enzymes promoting ether lipid synthesis, a recently described process promoting ferroptosis sensitivity (Zou et al. nature 2020) were also specifically increased in non-NE SCLC stickers (Fig. 4f). Importantly, we find that indeed non-NE SCLC is more sensitive to ferroptosis due to elevated ether lipid synthesis and as such becomes more resistant to ferroptosis upon suppression of AGPATG3 and AGPAT2 (Fig. 4g, h) and presents with elevated levels of ether lipid species (Fig. 4d, e). Thereby, we have now identified ether lipid plasticity to contribute to differential ferroptosis sensitivity of non-NE and NE SCLC subtypes, an additional mechanistic finding presented in whole new figure 4.

7. For Fig. 4F, in case of untreated, if possible select a different field/spot of cells.

This has been done and is now included in new Figure 5f. Of note, floaters will move a bit in the well during time-lapse experiments

8. Does BSO treatment also induce lipid ROS in ferroptosis sensitive SCLC?

This is an important question. To address this, we have now treated all three newly generated stickers with BSO and determined lipid ROS accumulation. Indeed, lipid ROS specifically accumulated upon treatment with BSO. Albeit at varying intensities in the three generated stickers. Importantly, in all cases, lipid ROS accumulation was prevented by Fer-1 co-treatment. These data are now added in Supplementary Fig. 7l.

9. The combination treatment of BSO/Auranofin seems to work well in SCLC (Fig. S7E,F), however, H82 cells which are sensitive to ferroptosis respond to BSO treatment and show a slight synergistic increase with Auranofin. On the other hand, H2171 cells resistant to ferroptosis, should be sensitive to Auranofin treatment alone but it doesn't show such behavior. The combination works very well but again from the data it is not clear what type of cell death is contributing to H2171 death. Inhibitors (Fer-1 and NAC) used are known to protect ferroptosis. Same thing is observed in Fig. S7C, Ferrostatin 1 alone doesn't rescue RP 250.1 cells but does so in combination with NAC,

We apologize for not mentioning this previously in the text, the experiments mentioned above using H82 and H2171 cells were done using sublethal doses of single drug treatment if possible to reveal their synergistic activity. For clarity we have mentioned this now in the text and have performed cross titrations in murine and human SCLC non-NE and NE cells to reveal the extent of synergy (Supplementary Fig. 8b, c). In both cases combined treatment improves single treatment activity of either BSO or Auranofin. We observe that cell death induced by the combination is blocked by combining Fer-1 and NAC, the latter blocking ferroptosis very poorly while being more potent at blocking non-lipid types of ROS (see also Dixon et al. 2012 and Prieto Clemente et al. 2020). Therefore, we conclude that the combination in H2171

induces a mixed type of ferroptosis and general ROS-driven death, the first being induced by BSO and the latter by Auranofin.

10. Ferrostatin 1 normally works at low concentrations (0.1 -0.4 μM), but the authors are using 5 μM , why?

As we initially started out using 5 μM we have, for consistency throughout the manuscript, performed all further experiments using this dose. We have, however now also included a titration of a range of doses for Ferrostatin-1 which all efficiently block ferroptosis as this reviewer rightly points out. These data are now added in Supplementary Fig. 2b.

11. In the lipidomics analysis, how did the authors differentiate between Plasmalogen + 1(O) and normal PE.

We performed a targeted analysis of selected oxidized phosphatidylethanolamine (PE) species as described in Doll et al., Nature 2019, 575, 693. The oxidized species were monitored using a triple quadrupole (QQQ) mass spectrometer (QTRAP 6500 LC/MS/MS System, SCIEX) on the basis of their specific Multiple Reaction Monitoring (MRM) transitions. In the MRM analyses, specific precursor ions - in our case the molecular ions ($[M-H]^-$) of the different PE species - were selected in the first quadrupole (Q1). These PE species were fragmented in the second quadrupole (Q2), which serves as collision cell. Simultaneously, the third quadrupole (Q3) was set to let pass only a characteristic PE fragment ion. The advantage of MRM analyses is that only ions, which fit both, the selected precursor and the characteristic fragment ions, can reach the detector whereas all other (lipid) species in the LC eluate are ignored.

To enable the selective detection of only those PE species, which are acylated by oxidized fatty acids, and the differentiation of these oxidized from non-oxidized PE species (including plasmalogens), we chose the oxidized fatty acyl chains as characteristic fragment ions in Q3.

To specifically answer the reviewer's question, we would like to give an example:

The m/z value of the $[M-H]^-$ precursor ion of the oxidized "Plasmalogen + 1(O)" species PE (16:0p/22:4(O)) is 766.5 Da. The $[M-H]^-$ precursor ion of the corresponding "normal PE" species PE (16:0/22:4) has the identical m/z value of 766.5 Da. This means that it is impossible to distinguish these two species only on the basis of their precursor ions. However, to be able to selectively detect only the oxidized PE (16:0p/22:4(O)), we used a MRM transition, for which we selected m/z 766.5 as precursor ion in Q1 and m/z 347.3 of the oxidized fatty acyl fragment $[22:4(O)-H]^-$ in Q3.

A short version of this explanation has been added to the methodology section.

REVIEWERS' COMMENTS

Reviewer #1 (Remarks to the Author):

The manuscript has addressed the issues of the initial review.

Reviewer #2 (Remarks to the Author):

The authors have addressed the comments from this reviewer sufficiently, and the current version of manuscript is much stronger and improved. This reviewer only has one more recommendation: the authors should carefully go through the manuscript and cite relevant references, some of which they have omitted. For example, the role of YAP1 and mitochondrial OXPHOS/ETC in ferroptosis have been reported (PMID: 31341276; PMID: 30581146), and these original reports should be cited in the manuscript.

Reviewer #3 (Remarks to the Author):

I want to congratulate the authors on their efforts and the amount of work done, all of my concerns have been answered. This manuscript has improved significantly.

Minor comments:

1. This is regarding my previous comment # 3, I do agree with authors that this is more transparent way of presenting the data. However, have the authors tried any other cell death/viability assay in these samples.
2. For Fig. 2A, crystal violet staining for cell line 250, Ferrostatin 1 doesn't seem to protect in erastin sample. An explanation or a different experiment may be helpful.

Point-by-point-response

Reviewer #1 (Remarks to the Author):

The manuscript has addressed the issues of the initial review.

Reviewer #2 (Remarks to the Author):

The authors have addressed the comments from this reviewer sufficiently, and the current version of manuscript is much stronger and improved. This reviewer only has one more recommendation: the authors should carefully go through the manuscript and cite relevant references, some of which they have omitted. For example, the role of YAP1 and mitochondrial OXPHOS/ETC in ferroptosis have been reported (PMID: 31341276; PMID: 30581146), and these original reports should be cited in the manuscript.

We apologize for not mentioning these citations as we are of course aware of them and they are indeed very relevant to our work. This has been corrected in the revised version.

Reviewer #3 (Remarks to the Author):

I want to congratulate the authors on their efforts and the amount of work done, all of my concerns have been answered. This manuscript has improved significantly.

We very much appreciate that this reviewer values our efforts.

Minor comments:

1. This is regarding my previous comment # 3, I do agree with authors that this is more transparent way of presenting the data. However, have the authors tried any other cell death/viability assay in these samples.

Indeed, we have and have previously not included these data for reasons of conciseness. To address this point however, we have now included these viability assays for erastin, RSL3 and ML210 treatment +/- Fer-1 in H82 cells in revised supplementary figure 2b. The data confirm our previous conclusions that all three treatments induce cell death blockable by Ferrostatin-1.

2. For Fig. 2A, crystal violet staining for cell line 250, Ferrostatin 1 doesn't seem to protect in erastin sample. An explanation or a different experiment may be helpful.

We agree, that erastin-induced suppression of clonal outgrowth is not entirely blocked by Fer-1 on all three cell lines but most evidently in 250. This is however expected, as erastin is known to not only induce ferroptosis but to also induce other types of cell death. The use of Fer-1 highlights to which extent erastin induces ferroptotic cell death in these cells and which effects are unrelated to that and likely a result of effects on other pathways in this long-term assay. We have included a sentence clarifying this point in the revised manuscript.